


**Transparent exopolymer particle binding of organic and inorganic**
**particles in the Red Sea: Implications for downward transport of**
**biogenic materials**
**Abdullah H. A. Dehwah[1,6], Donald M. Anderson[2], Sheng Li[1,4], Francis L. Mallon[3],**
**Zenon Batang[3], Abdullah H. Alshahri[1], Michael Hegy[4], Thomas M. Missimer[5]**
[1] King Abdullah University of Science and Technology (KAUST), Water Desalination and Reuse
Center (WDRC), Biological and Environmental Science and Engineering (BESE), Thuwal 23955-
6900, Saudi Arabia
[2]Woods Hole Oceanographic Institution, Biology Department, Woods Hole, MA 02543, USA
[3]Coastal and Marine Resources Core Laboratory, King Abdullah University of Science and
Technology (KAUST), Thuwal, Saudi Arabia
[4]Guangzhou Institute of Advanced Technology, CAS, Haibin Road #1121, Nansha district,
Guangzhou 511458, China
[5]U. A. Whitaker College of Engineering, Emergent Technologies Institute, Florida Gulf Coast
University, 16301 Innovation Lane, Fort Myers, Florida 33965-6565
[6]Desalination Technologies Research Institute (DTRI), Saline Water Conversion Corporation
(SWCC), P.O. Box 8328, Al-Jubail 31951, Saudi Arabia





**Abstract:** Binding of particulate and dissolved organic matter in the water column by marine
gels allows sinking and cycling of organic matter into deeper water of the Red Sea and other
marine water bodies. A series of four offshore profiles were made at which concentrations of
bacteria, algae, particulate transparent exopolymer particles (p-TEP), colloidal transparent
exopolymer particles (c-TEP), and the fractions of natural organic matter (NOM), including
biopolymers, humic substances, low molecular weight neutrals, and low molecular weight acids
were measured to depths ranging from 90 to 300 m. It was found that a statistically-significant
relationship occurs between the concentrations of p-TEP and bacteria while a minimal, non-
significant relationship between p-TEP and algae occurs. This likely reflects the low abundance
of larger algal species in the study region. Variation in the biopolymer fraction of NOM in
relationship to TEP and bacteria suggests that extracellular discharges of polysaccharides and
proteins from the bacteria and algae are occurring without immediate abiotic assembly into p-
TEP. In the water column below the photic zone, TOC, bacteria, and biopolymers show a
generally common rate of reduction in concentration, but p-TEP decreases at a diminished rate,
showing that it persists in moving organic carbon deeper into the water column despite
consumption by bacteria.
________________________________________________________________________________
**1 Introduction**

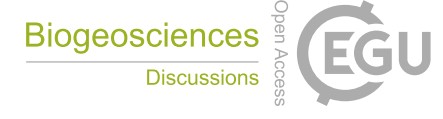

Mechanisms that control the biogeochemical cycles influenced by microorganisms in the world
's oceans are complex and poorly understood (Azam and Malfatti, 2007). The relationships
between microalgal and bacterial abundance, total organic carbon (TOC), fractions of natural
organic matter (NOM), polysaccharides, and transparent exopolymer particles (TEP) in seawater
with depth play important roles in the transport and cycling of nutrients and sediments
(Alldredge and Crocker, 1995; Passow, 2002; Azam and Malfatti, 2007). In particular, the
binding of suspended sediments and particulate organic matter by TEP and other acidic
polysaccharides, in addition to general aggradation, tends to increase particle size and weight,
thus increasing settling rates in the water column (Passow et al., 2001; Wurl et al., 2011).  It has
been demonstrated that gel-type particles link particulate and dissolved organic matter in the
ocean (Verdugo et al., 2004). The sinking of biogenic particles drives elemental cycling, which in
turn controls primary and secondary productivity through the water column (Wurl et al., 2011).
Particulate organic material is commonly occupied or influenced by bacteria which can reduce
the biomass by consumption of some organic matter over various timeframes from days to
weeks (Bižić-Ionescu et al., 2018).
TEP are ubiquitous in the oceans (Passow, 2002), likely caused by abiotic coagulation and
aggradation of dissolved carbohydrates or primarily acidic polysaccharides, but also by biotic
formation as extracellular secretions by algae or bacteria (Chin et al., 1998; Stoderegger and
Herndl, 2001; Passow et al., 2001; Berman and Viner-Mozzini, 2001; Passow et al., 1994).
Particulate TEP (p-TEP) is in the size range 0.4–200 µm, with a number of forms, including
amorphous blobs, disseminated clouds, sheets, filaments or clumps (Zhou et al., 1998; Passow,
2002; Mari et al., 2004). Colloidal TEP (c-TEP) consist of particles that are stained by Alcian blue





with a diameter range of 0.05 to 0.4 µm (Villacorte al., 2009). However, c-TEP is defined based
solely on staining with Alcian blue, which is known to also stain other substances in seawater,
including sulfated and carboxylated polysaccharides, glycoproteins, polyanions in general, and
acidic polysaccharides not associated with TEP (Winters et al., 2016).

TEP are composed of acidic polysaccharides enriched with fucose and rhamnose, thus

serving as a food source in the water column and commonly associated with layers of intense
microbial and biochemical activity (Azam and Long, 2001). TEP generally decrease in
concentration with depth in the sea (Engel et al., 2004), with a tendency to float to the sea
surface if unballasted to contribute a gelatinous layer to the sea surface microlayer (Azetsu-
Scott and Passow, 2004; Wurl and Holmes, 2008; Wurl et al., 2009). Bar-Zeev et al. (2015) have
documented that p-TEP is mainly composed of polysaccharides, which can be dispersed in the
presence of different types of chelators, be fractured to form colloids or reassemble abiotically.
The trends in TEP concentrations in the seawater column have been previously examined
(Jennings et al., 2017), and the relationship of TEP with TOC, DOC, and bacteria have also been
investigated in many areas of the ocean (Engel, 2004; Simon et al., 2002; Ortega-Retuerta  et al,
2009; Ortega-Retuerta  et al.,  2011; Bar-Zeev et al., 2011). However, these relationships have
not been studied in the Red Sea.

Studies on TEP distribution in relation to other forms of organic matter in the Red Sea have

focused mainly on assessing the links between TEP and phytoplankton and bacterial production
(Bar-Zeev et al., 2009b) and the impacts of TEP and dissolved forms of NOM on biofouling in
seawater desalination plants (Bar-Zeev et al., 2009a; Dehwah et al., 2015a; Dehwah et al.,
2015b; Dehwah et al., 2015c; Dehwah and Missimer, 2016; Rachman et al., 2014; Rachman et




al., 2015). The intakes for reverse osmosis seawater desalination plants are located in shallow,
nearshore areas of the Red Sea, so little consideration has been given to changes in TEP
concentration with depth until it was suggested that deep-water intake systems may produce
seawater quality with lower concentrations of algae, bacteria, and organic compounds, such as
TEP, thus possibly lessening rates of membrane biofouling (Dehwah et al., 2015c).

The relationships between TEP concentrations and abundance of microalgae, bacteria, TOC,

and dissolved fractions of NOM, including biopolymers, humic substances, building blocks, low
molecular weight (LMW) acids, and LMW neutrals from the sea surface to 300 m depth are
herein presented.

The present study provides the first data from the Red Sea, with initial insights into the

vertical transport of organic carbon, including the fractions of natural organic matter from the
surface to depths near or below the photic zone. The authors are keenly aware that the data
presented herein have not been collected in a systematic manner with spatial and temporal
comparisons to assess the biogeochemical cycles within the Red Sea comprehensively.
However, the compiled data can be used to better characterize the biogeochemical cycles of
the Red Sea as other researchers add new data. The reported datasets represent the first
measured in the Red Sea wherein the fractions of organic matter, including biopolymers, humic
substances, building blocks, low molecular weight neutrals, and low molecular weight acids
(very expensive to measure), are linked with measurements of algae, bacteria, TOC, and TEP.

**2 Methods**





2.1 Compilation and comparison of available data

There have been several investigations on organic matter, including TEP, collected at depths
near the sea surface along the Red Sea coast of Saudi Arabia, with the main focus to establish
the relationships between seawater organic matter content and the potential for membrane
biofouling in seawater desalination facilities. These shallow nearshore data were compiled and
assessed to compare to the newly collected offshore data and to assess statistical relationships
between various organic parameters. Note that these data have been collected at many
different times of the year and were not used to attempt the characterize the natural seasonal
variations and the overall biochemical activity in the nearshore area of the Red Sea.

2.2 Seawater vertical profiles in the Red Sea

Seawater properties of the water column were measured at four sites (A–D) north of Jeddah,
along the Saudi coast of the Red Sea in deep water areas (> 1000 m) (Fig 1). In situ vertical
profiles of temperature, salinity, dissolved oxygen (DO), pH, turbidity, chlorophyll-a
(fluorescence), and photosynthetically active radiation (PAR) were determined with a multi-
sensor assembly fitted to a Rosette carousel holding a set of Niskin water sampling bottles
(General Oceanics, USA). Continuous vertical profiling was conducted from sea surface to 90 m
depth at sites A–C, with seawater samples obtained at 10 m depth intervals for the analysis of
organic parameters. At site D, continuous vertical profiles of physicochemical parameters were
taken from 7 m below sea surface to 300 m depth, with seawater samples for organic





parameters obtained at 10 m intervals from the surface to 100 m depth and at 20 m intervals
thereafter to 300 m depth. Sampling at sites A–C was conducted in April 2014, whereas at site
D in February 2015. The sample timing was based on ship availability and the data collected
cannot be used to fully characterize the Red Sea in deep water located far from the coast. Note
that the water depth drops almost vertically to greater than 1000 m beginning in the nearshore
at the 20 m contour (Dehwah et al., 2015c).

The multi-sensor assembly included the SBE 43 CTD device (Sea-Bird Scientific) for salinity,

temperature and depth profiling, with a DO add-on sensor; Wet Labs ECO AFL/FL (Sea-Bird
Scientific) was used for turbidity and fluorescence detection; and a biospherical light sensor (LI-
COR) was used for PAR measurement. All sensors were pre-calibrated according to
manufacturer specifications before actual use in field sampling and was normalized.

2.3 Quantification and characterization of microalgae and bacteria

Microalgal abundances in water samples were determined by flow cytometry, using a BD
FACSVerse flow cytometer for counting and characterizing algal cells. An Accuri flow cytometer
was used to measure bacterial abundance. Flow cytometry enables a rapid and accurate
counting of microorganisms (VivesRego et al., 2000).

Light scattering properties and/or fluorescence intensity was determined by the flow

cytometer to distinguish between different algal types, as described by van der Merwe et
al. (2014). Lasers were used to excite both unstained autofluorescent organisms (algae) and
stained bacterial cells. The red laser wavelength was set at 640 nm and the blue laser at 488 nm





for the Accuri flow cytometer. Algal cell counting was performed by combining 500 μL of each
sample with a 1 μL volume of a standard containing 1 μm beads to calibrate size in a 10 mL
tube. The tube was then vortexed and measured at high flow rate with a 200 μL injection
volume for 2 min. The counting procedure was repeated three times to assess the precision of
measurements. There different type of algae, cyanobacteria, *Prochlorococcus*, and
pico/nanoplankton, were distinguished based on their autofluorescence as well as by the cell
side-angle scatter, which was used to identify them by size (Radíc et al., 2009).

A comparative protocol employing SYBR®Green stain was used for bacteria counting. A

volume of 500 μL from each sample was transferred to a 10 mL tube, incubated in 35°C water
bath for 10 min. SYBR® Green dye was added at a 5 μL into a 500 μL aliquot to stain the cells.
The sample was vortexed and incubated for 10 min. The prepared samples were then analyzed
at a medium flow setting with a 50 μL injection volume for 1 min. For validation, 8-Peak
calibration beads were used. Triplicate measurements were made on each sample to assess
measurement precision.

2.4 Measurement of TOC and NOM fractions

TOC concentration was measured with a Shimadzu TOC-VCSH. Fractions of dissolved organic
carbon, including biopolymers, humic substances, building blocks, low molecular weight (LMW)
neutrals, and low molecular weight (LMW) acids, were determined by Liquid Chromatography
Organic Carbon Detector (LCOCD, DOC-Labor), using a size exclusion chromatography column
Toyopearl HW-50S (TOSOH), following the methods by Huber et al. (2011). A calibration curve





was established for both molecular masses of humic substances and detector sensitivity before
sample measurements. Humic acid and fulvic acid standards (Suwannee River Standard II) were
used for the molecular mass calibration, whereas potassium hydrogen phthalate and potassium
nitrate ($KNO_3$) for sensitivity calibration based on Huber et al. (2011).

All seawater samples for LCOCD were manually pre-filtered using a 0.45 µm syringe filter to

exclude the undissolved particulate organics. Before sample analysis, a system cleaning was
performed by injection of 4,000 µL of 0.1mol/L NaOH through the column for 260 min. After
cleaning, 2,000 µL of the sample was injected for analysis at 180 min retention time and 1.5
mL/min flow rate. A mobile phase of phosphate buffer, with 28 mmol STD and 6.58 pH, was
used to carry the sample through the system. The resulting chromatogram showed a plot of
signal response of different organic fractions against retention time. Manual integration of the
data, also following Huber et al. (2011), was performed to determine the concentrations of the
different organic fractions, including biopolymers, humic substances, building blocks, LMW
acids and LMW neutrals.

2.5 TEP measurement

Both p-TEP and c-TEP were simultaneously determined in each collected sample. The size range
of p-TEP is between 0.4 and 200 µm, whereas c-TEP between 0.05 and 0.40 µm (Villacorte et
al., 2009). TEP analysis was based on the method developed by Passow and Alldredge (1995),
which involves sample filtration, membrane staining with Alcian blue, and then UV
spectrometry. A staining solution was prepared from 0.06% (m/v) Alcian blue 8GX (Fluka) in



acetate buffer solution (pH 4) and freshly pre-filtered through a 0.2 μm polycarbonate filter
before usage. A 300 mL volume of seawater from each water sample was filtered through a 0.4
μm pore size polycarbonate membrane using an adjustable vacuum pump at low constant
vacuum. After filtration, the membrane was rinsed with 10 mL of Milli-Q water to prevent the
Alcian blue from coagulating, as salts may remain on the filter after seawater filtration, thus
avoiding the likelihood of overestimating the TEP concentration. The retained TEP particles on
the membrane surface were then stained with the Alcian blue dye for 10 seconds. After
staining, the membrane was flushed with 10 mL of Milli-Q water to remove any excess dye. The
flushed membrane was then placed into a small beaker, where it was soaked in 80% sulfuric
acid for 6 hours to extract the dye that was bound to the p-TEP. Finally, the absorbance of the
acid solution was measured by a UV spectrometer at 752 nm wavelength to determine the TEP
concentration. The same methodology was applied to determine the colloidal TEP, except that
a 250 ml volume of water sample from 0.4 μm polycarbonate membrane permeate was filtered
through a 0.1 μm pore size to allow deposition of the c-TEP on the membrane surface.

To relate the measured UV absorbance values to TEP concentrations, a calibration curve

was established. Xanthan gum solutions with different volumes (0, 0.5, 1, 2, 3 mL) were used to
obtain the calibration curve (Fig. 2). The TOC concentrations of xanthan gum before and after
0.4 μm filtration were analyzed, and the TOC concentration difference was used to calculate
the gum mass on each filter and the TEP concentration was estimated using the calibration
curve. The same procedures were used for the 0.1 μm membrane to establish the calibration
curve for colloidal particles. Afterwards, the TEP concentration was expressed in terms of
Xanthan Gum equivalent in μg Xeq./L by dividing the TEP mass by the corresponding volume of





TEP samples. Because particulate and colloidal TEP is determined indirectly, these values must
be considered to be semi-quantitative. The new method developed by Villacorte et al. (2009)
for TEP measurement was not used, as it would limit the comparability of the measured data
with previous results.

2.6 Statistical methods used for data comparison
A series of scatter plots were constructed to test the statistical significance between various
organic parameters. The $R^2$ and p-values were calculated to assess the degree of fit to a curve
and the statistical significance. When the p value was below 0.05, the null hypothesis was void
and the relationship was deemed to be significant.

**3 Results**

3.1 Variations in salinity, temperature, fluorescence, pH, dissolved oxygen, PAR/irradiance,

biospherical/licor, and turbidity


The thermocline in the three profiles (sites A-C) collected to a 90 m depth showed a slight
decrease in temperature from near 29 °C to between 24 and 25 °C 90 m below surface (Fig. 3).
The decline in temperature was relatively gradual at all three sites. In the deep profile, the
temperature declined from about 26.5 °C at the surface to about 22 °C at 300 m. An inflection
point occurred at about 115 m and the change in temperature below this depth to 300 m was
only about 2.5 °C (Fig. 4). The difference in the temperature at the sea surface between profiles





was likely caused by the time of year of measurements, with the 90 m profiles occurring in April
versus the 300 m profile in February which is the peak of winter in the study area.
The halocline showed similar salinity variations in the 90 m profiles with a slight, rather
uniform increase from about 39 ppt at surface to 40 ppt at 90 m (Fig. 3). A slightly lower salinity
gradient coinciding with a slightly higher temperature gradient occurred at site B. The salinity
change in the 300 m profile showed a similar pattern from about 39 to 40 ppt in the upper 115
m, but an inflection occurred at about 115 m wherein the rate of increase declined to a few
tenths of a ppt over the lower 185 m. The inflection point showing a slope change for both
temperature and salinity occurred at about the same depth (Figs. 3 and 4).
The vertical trends in pH also exhibited minimal variations down to 90 m at sites A–C (Fig.
3), but with slightly lower pH values at site A (7.9–8.0) than at sites B and C (8.0–8.1). In the
deep profile, pH was nearly stable at about 8.3 until 115 m and then steadily decreased to 8.1
at 300 m (Fig. 4).
Dissolved oxygen (DO) concentrations in the shallow profiles at all three sites showed high
variability (6–12.5 mg/L) in the top layer (unknown reason for variation), but with relative
stability at about 5 mg/L from 20 to 90 m (Fig. 3). DO in the deep profile was at lower
concentrations (0.8–1.5 mg/L) near the surface, increasing to around 2 mg/L at 115 m and then
steadily declined to about 0.6 mg/L at 300 m with a saturation of only 10%.
The vertical pattern in chlorophyll $a$ (chl-$a$) concentrations markedly differed between
shallow and deep profiles (Figs. 3 and 4). At sites A–C, chl-$a$ was slightly detected at the surface
but abruptly increased from 0.3 to 1.2 mg/m$^3$ within 50–75 m and thereafter declined to near
0.2 mg/m$^3$ at sites A and C and to about 0.5 mg/m$^3$ at site B. Chl-$a$ concentrations were

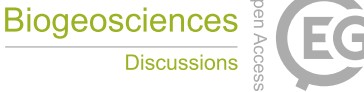



relatively low in the deep profile, decreasing from about 0.45 mg/m$^3$ at the surface to about
0.06 mg/m$^3$ at 100 m, from which it remained unchanged until 300 m. Note that these chl-*a*
concentrations were based on *in situ* fluorescence detection using a sensor that was pre-
calibrated with a chlorophyll standard from the manufacturer (Wet Labs). As chlorophyll
fluorescence may vary with cell physiological condition, time of day, light regime, and other
factors, and since the sensor was not field-validated after calibration, the present chl-*a* values
should thus be considered semi-quantitative.

PAR levels at sites B and C were initially recorded at 600-700 µmol/m$^2$/s at the surface and

then steeply decreased to 120-160 µmol/m$^2$/s at 20 m depth, from where it further decreased
gradually until 90 m depth (Fig. 3). At site A, where the measurement was done at an earlier
time, PAR varied between 220-300 µmol/m$^2$/s within the top 10 m layer and then coincided
with the same values at sites B and C. PAR in the deep profile steeply declined from about 240
µmol/m$^2$/s near the surface to about 20 µmol/m$^2$/s at 40 m depth, after which it gradually
decreased to near zero at about 75 m depth, which is generally similar to the trend in the
shallow profiles (Figs. 3 and 4). The depths at which the PAR levels were at 1% of the surface
values were in range of 38–54 m for all sites.

Turbidity was generally low in the vertical profiles at all sites. Turbidity varied in the narrow

range of 0.2-0.3 NTU, with only a few spikes up to 0.4 NTU, in all three shallow profiles (Fig. 3).
In the deep profile, most turbidity values were within 0.1–0.15 NTU, with intermittent spikes up
to 0.2 NTU below 75 m depth (Fig. 4).

3.2 Algae and cyanobacteria concentrations




Total concentrations of algae and cyanobacteria (summed) with depth at the shallow and deep
sampling stations are shown in Figs. 5 and 6, respectively. Previous results on total algal and
cyanobacterial abundances, all collected from surface layers close to shore in the same study
area, are compiled in Table 1, with a range of 1,677−137,363 cells/mL (mean 44,383 cells/mL
out of 38 samples). Total algal and cyanobacterial concentrations from the surface at the
shallow stations (A−C) during the present study were comparable to the mean of the previous
data, while the surface concentration at the deep station (D) was close to the reported
maximum (Figs. 5 and 6, Table 1).

The vertical profiles of algal and cyanobacteria concentrations by group (cyanobacteria,

*Prochlorococcus* and pico/nanoplankton) are shown in Figs. 5 and 6 for the shallow (A-C) and
deep sites, respectively. At sites A-C, cyanobacteria were more abundant near the surface (top
10 m layer), below that *Prochlorococcus* was more predominant, with peak concentrations at
about 50 m (Fig. 5). In general, algal and cyanobacterial concentrations showed a substantial
decline below 80 m at all sites. The same compositional and abundance trends were exhibited
in the deep profile, except that cyanobacteria had higher concentrations near the surface in the
deep profile while *Prochlorococcus* was relatively denser at subsurface depths in the shallow
profiles (Fig. 6). In addition, the concentrations of pico/nanoplankton in the upper layers were
relatively higher at the deep site compared to the shallow sites (A-C) (Figs. 5 and 6).

3.3 Bacteria concentrations





The vertical trends in bacterial concentrations during the present study are shown in Figs. 7 and
8, indicating higher cell densities in the upper 50 m layer at the deep site compared to sites
A–C. Previous results on nearshore bacterial concentrations from the same study area ranged
from $1.13 \times 10^5$ to $2.18 \times 10^6$ cells mL$^{-1}$ (mean $5.26 \times 10^5$ cells mL$^{-1}$; 40 samples) (Table 1). The
new data on offshore surface concentrations of bacteria are comparable to the average of the
nearshore results (Table 1, Figs. 7 and 8). Bacterial abundance generally declined with depth,
with a decrement of about $4.00 \times 10^5$ to $9.00 \times 10^4$ cells mL$^{-1}$ from the surface to 90 m depth at
sites A–C (Fig. 7) and from about $5.00 \times 10^5$ cells mL$^{-1}$ at the surface to $1.60 \times 10^5$ cells mL$^{-1}$ at
160 m and to $1.00 \times 10^5$ cells mL$^{-1}$ at 300 m (Fig. 8).

3.4 Total organic carbon (TOC)

TOC concentrations exhibited only minor differences between the sites, with fluctuations
within a narrow range in the upper 50 m layer at both the shallow and deep sites (Figs. 7 and 8).
Nearshore data on TOC ranged from 0.83 to 1.42 mg/L, with an average of 1.0 mg/L from 42
measurements (Table 1). In the offshore near-surface profiles, the TOC ranged from 0.99 to
1.35 mg/L.
TOC generally declined with depth at all sites, although only within a narrow range at sites
A–C between 1.2 mg/L at surface and 0.9 mg/L at 90 m depth. The decline in the deeper profile
was from 1.1 mg/L at surface to 1.0 mg/L at 120 m and then to 0.75 mg/L at 300 m.

3.5 Particulate and colloidal TEP concentrations






Nearshore p-TEP and c-TEP showed considerable variation in concentrations with ranges of
53−347 (mean 191) and 36−287 (125) µg Xeq./L, respectively (Table 1). Comparable ranges of
concentration for both parameters were found offshore, except for the markedly higher c-TEP
concentrations in the vertical profile at site A (Figs. 7 and 8).

Both p-TEP and c-TEP generally declined with depth, although with fluctuations between

50−100 m depth and an elevated value at 200 m depth in the deep profile. The difference in
concentrations was more pronounced for c-TEP in the deep profile, from 265 µg Xeq./L at 10 m
to about 70 µg Xeq./L at 300 m. The change in concentration of p-TEP with depth in the deep
profile was relatively slight, from about 285 µg Xeq./L at 40 m to 170 µg Xeq./L at 300 m. At the
shallow sites, both p-TEP and c-TEP trends with depth showed similar patterns between sites B
and C, except that c-TEP was unusually low in the surface layer at site C (Fig. 7).

3.6 NOM fractions

There was considerable variability in concentrations of the NOM factions in nearshore seawater
The range in concentration, number of samples, and average of the concentrations are the
following: biopolymers (28-164 µg/L, 42, 62  µg/L), humic substances (159-442 µg/L, 42, 248
µg/L), building blocks (81-260 µg/L, 42, 118 µg/L), LMW neutrals (16-477 µg/L, 42, 271 µg/L),
and LMW acids (10-130 µg/L, 42, 40 µg/L). The range in biopolymer concentrations in the
surface offshore samples are similar to the nearshore samples. All of the NOM fractions have
higher concentrations at the A, B, and C profiles compared to the deep profile.



The biopolymer fraction of NOM shows a general reduction with depth in all offshore
profiles. At sites A and B there is a spike in biopolymers at 10 m with minor variation between
10 m to 90 m. In the deeper profile, there is considerable variation in the photic zone with the
surface having the highest value and subsequent spikes occurring at 30 and 60 m. Beginning at
about 90 m, there is a constant downward trend in concentration.
Humic acid concentrations showed only minor variations with depth in the shallow profiles,
but the deep profile showed a reduction by about 29% from 90 to 300 m depth. There is a
general decreasing trend in concentration of building blocks with depth at the deep site and
only minimal differences throughout the depth profiles at sites A–C (Figs. 7 and 8). The
concentrations of LMW neutrals at the shallow sites were the highest amongst NOM fractions,
although with a wide range of variation. In contrast, LMW acids had the lowest concentrations
without marked discrepancies in concentration in the vertical profiles between sites A–C, but a
general reduction occurred below 120 m in the deep profile (Figs. 7 and 8).

**4 Discussion**

4.1 Algal and cyanobacterial concentrations

The flow cytometry approach used in this study was highly effective in characterizing and
enumerating the small size classes of phytoplankton and cyanobacteria that are readily
distinguishable on the basis of cell size and autofluorescence.  Thus, cyanobacteria (presumably
*Synechococcus* spp), *Prochlorococcus*, and the general class of pico/nanoplankton were





numerically dominant, with very few larger eukaryotic algal species detected. This is consistent
with prior studies that reported that phytoplankton in the oligotrophic northern Red Sea and
Gulf of Aqaba are dominated (>95%) by cells <5 µm in size (Lindell and Post 1995; Yahel et al.
1998). Only during the summer does the large macroalgae *Trichodesmium* sp. also become
prominent. As reported here, algae ranging from 5 to several hundred µm are extremely scarce,
although not totally absent (Sommer 2000; Kimor and Goldanski 1992).

4.2 Correlations between TEP, bacteria, algae, the biopolymer fraction of NOM, and TOC

TEP is composed of acidic polysaccharides and some large proteins that occur mostly in the
biopolymer fraction of NOM and some of the proteins within the humic acid part of NOM (Bar-
Zeev et al. 2015; Winters et al. 2016). TEP can be produced both abiotically and as extracellular
discharges from bacteria and algae (Zhou et al. 1998; Passow et al. 2001; Passow 2002; Engel et
al. 2004; Iuculano et al. 2017). Therefore, there should be some statistical relationship between
TEP, the biopolymer fraction of NOM, bacterial concentration or algal concentration.
A series of statistical analyses were preformed to test if there are significant relationships
between the various organic properties (Table 2). In all cases there was no statistically-
significant relationship between any of these parameters in the shallow, nearshore samples
with the exception of bacteria and TOC. However, some important and statistically-significant
relations were found between p-TEP and bacteria in the profiles measured at sites A, B, and C
and in the 300 m profile. All of the offshore profiles showed a statistically significant
relationship between c-TEP and bacteria with the exception of the site C profile. In comparison,



there was only one statically-significant relationship between p-TEP and algae at site B and for
c-TEP the only profile showing statistical-significance was the deep profile. Based on these
relationships, it appears that p-TEP may be produced by bacteria in greater amounts compared
to algae at these locations. Consumption of the TEP by bacteria does not seem to be occurring
in the water column at sites A, B, and C based on the p-TEP relationship, unless abiotically-
generated p-TEP is replacing what is consumed. In the deep profile, the c-TEP concentration
shows a statistically-significant relationship with bacteria which could indicate a breakdown of
the p-TEP, particularly outside of the photic zone.
The relationship between the biopolymers and bacterial concentrations shows a significant
statistical correlation in all offshore profiles while the correlation between biopolymers and
algal concentrations is statistically significant only at site A and in the deep profile. These
relationships suggest that extracellular discharges of polysaccharides and proteins from the
bacteria and algae are occurring without immediate abiotic assembly into p-TEP. This
suggestion is further supported by the statistical relationships between biopolymers and p-TEP
and c-TEP which are statistically significant at several, but not all of the offshore profiles.
The offshore profiles show statistically-significant relationships between both p-TEP and c-
TEP and TOC at sites A and the deep profile. Therefore, TEP in general is a significant part of
TOC in the Red Sea at these locations, particularly below the photic zone.
There is usually no statistical relationship of significance between p-TEP and c-TEP.
However, at site A the data produced an $r^2$ value of 1 and with a corresponding p-value of 0
(Table 2). This unusual relationship has no explanation but is noted.

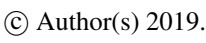


A considerable amount of additional research will be required to better establish the
processes occurring within the Red Sea water column that relate to NOM production and
transport and how these processes relate to the measured TEP and NOM fraction
concentrations. Since there are few data available in the literature that relate these parameters
within the water column at other geographic locations, it is difficult to provide definitive
conclusions. The data provided here appear to be the first published that relate the biopolymer
fraction of NOM to TEP and provide all of the five fractions of NOM in the offshore marine
environment throughout the water column. The carbon compounds that occur in p-TEP are
largely contained within the biopolymer fraction of NOM with the exception of some proteins
which occur in the size range found in the humic substances.
An assessment of the other fractions of NOM, humic substances, building blocks, LWM
neutrals, and LMW acids, did not show any significant statistical relationships between these
parameters, nor did it reveal potential relationships between them and the bacteria or algae.

4.3 Comparison of the offshore and onshore TEP data in the Red Sea

All of the onshore measurements of p-TEP and c-TEP were collected between the sea surface
and a depth of 10 m. Therefore, only the data in this depth range can be compared to the
offshore data. The full range of p-TEP in the nearshore measurements is from 53 to 347 µg
Xeq./L and the c-TEP range is between 36 and 287 µg Xeq./L. The ranges in the offshore profiles
in the same depth range for p-TEP and c-TEP are 135.4 to 279.4 and 0 to 340.7 respectively. In
both locations there was considerable variation between sites and in different times of the year



which is expected based on production variations of TEP by algae and bacteria in the upper
photic zone as well as the ability of TEP to have either negative or positive buoyancy at shallow
depths (Zhou et al., 1998; Passow, 2002; Mari et al., 2004; Schuster and Herndl 1995; Ortega-
Retuerta et al. 2017).

4.4 Comparison of TEP profiles in the Red Sea with other marine environments

Most TEP data profiles collected in the marine environment show an irregular variation in the
upper 100 m of the water column (Schuster and Herndl 1995; Ortega-Retuerta et al. 2017), a
general reduction of TEP with depth over 200 m (Busch et al. 2017; Jennings et al. 2017), but in
some cases an increase at greater depths (Ramaiah et al., 2000). Also, the reported changes in
TEP with depth are based mostly on p-TEP data and not both types of TEP which show differing
trends in the water column. The TEP data collected from the profiles in this investigation within
the photic zone (<100m) show differing concentrations with depth (Figs. 7 and 8). Within the
upper 90, p-TEP declines between 31 and 39% at sites A, B, and C and shows no decline in the
deep profile. The c-TEP concentration declines between 38 and 70% at sites A, B, and the deep
profile, but increases by 150% at site C. For comparison, the TOC concentration reduction in the
photic zone ranges between 10 and 32%. In the deep profile the difference between the
surface and the 300 m depth showed a reduction in p-TEP of 20% and c-TEP of 69%. This may
indicate that some abiotic assembly of p-TEP is occurring below the photic zone, particularly in
the presence of bacteria which may feed upon the p-TEP. The TOC in the deep profile declines
by about 32% comparing the surface to the 300 m depth.




## 4.5 Relationships between NOM fractions and other parameters


The primary fraction of NOM that shows a trend with depth is the biopolymers which track
well to bacteria. Since the biopolymer fraction of NOM contains most of the polysaccharides,
which are food for bacteria, the relationship with the bacteria is to be expected. In the upper
100 m of the water column, the humic substances show a restricted range in concentrations
with a small downward trend (Fig. 7), but below 100 m there is a lowering concentration
following the same pattern as the biopolymers. The building blocks have a larger range in
concentration changes in the upper 100 m of the water column compared to the humic
substances (Fig. 7) and a similar downward trend in concentration similar to the humic
substances below 100 m (Fig. 8). The LMW neutrals and acids show considerable variation in
concentration in the upper 100 m and a slight downward change in concentration below 100 m.
There are some general suggestions made by these data related to the concentration
changes. In the photic zone, the biochemical activity of algae and bacteria affect the NOM
fraction concentrations. The LMW fractions are likely affected by the biochemical breakdown of
large molecular weight organics and by selective, abiotic aggradation of larger organic particles
suggested by the larger concentration of the neutrals over the acids. The reduction in
concentrations in biopolymers, humic substances and building blocks below 100 m follows the
reduction in bacteria below the photic zone. As bacteria feed on p-TEP, they may leave behind
the LMW neutrals which could be compounds that cannot be used by the bacteria as food. The
LMW acids may tend to occur within the context of c-TEP and may be subject to abiotic



aggradation during settling. Future research will be required to understand the complex
relations between the NOM organic fractions and the biochemistry of the bacteria in the deep-
water column.

**5 Conclusions**

Vertical changes in concentrations of TEP in the Red Sea tend to follow trends found in other
locations of the world ocean in that there is a general reduction with depth. The changes in the
photic zone tend to be quite irregular, as expected, because of variations in primary
productivity and differing biochemical conditions. Although it was observed that no clear
relationship between TEP and algae occurs in the Red Sea, this unusual result may be explained
by the dominance of small algae and cyanobacteria. The measurement of the five fractions of
NOM allows some preliminary conclusions to be made concerning the relationships between
specific organic parameters and TEP variation with depth. These relationships suggest that
extracellular discharges of polysaccharides and proteins from the bacteria and algae are
occurring without immediate abiotic assembly into p-TEP in the photic zone of the water
column. In the water column below the photic zone, TOC, bacteria, and biopolymers show a
generally common rate of reduction in concentration, but p-TEP concentration changes at a
reduced rate showing that it persists in moving organic carbon deeper into the water column
despite consumption by bacteria. There may be some abiotic assembly of c-TEP into p-TEP to
maintain the concentration without full bacterial removal.





The relationships between p-TEP and c-TEP and other organic parameters, especially the

biopolymer fraction of NOM, is different when comparing the offshore water column to the

nearshore area. The only statistically-significant relationship in the measured parameters in the

nearshore was that between bacteria and TOC. Irregularity in local conditions in the nearshore

zone causes large variations in the organic parameters measured, not allowing statistically-

significant relationships to be established.

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

Table 1. Compilation of related data from previous studies

| Location | Date | Depth (m) | Total Algae (cells/mL) | Bacteria (cells/mL) | TOC (mg/L) | NOM (µg/L) | | | |
|---|---|---|---|---|---|---|---|---|---|
| | | | | | | Biopoly. | Humic substances | Building Blocks | LMWN |
| [1]N. Obhor | 1/8/2014 | Surface | 30,524 | 112,790 | 0.89 | 76 | 345 | 103 | 168 |
| [1]Corniche | 1/11/2014 | Surface | 3,603 | 196,377 | 0.94 | 90 | 360 | 91 | 192 |
| [1]S. Jeddah | 1/9/2014 | Surface | 1,677 | 264,728 | 1.02 | 116 | 351 | 139 | 197 |
| [1]Buhayrat | - | Surface | 30,395 | 320,870 | 1.053 | 47 | 343 | 82 | 16 |
| [2]Site A (Buhayrat) | 1/7/2014 | Surface | 14,956 | 179,837 | 0.88 | 63 | 367 | 131 | 230 |
| [2]Site B (Saudia) | 5/25/2013 | Surface | 23,773 | 317,174 | 0.83 | 84 | 289 | 101 | 45 |
| [3]N. Obhor | 10/25/2014 | Surface | 129,738 | 520,350 | 1.1 | 57 | 205 | 95 | 163 |
| [3]Corniche | 11/6/2014 | Surface | 89,033 | 254,450 | 1.0 | 44 | 201 | 86 | 249 |
| [3]S. Jeddah | 12/24/2014 | Surface | 42,923 | 216,400 | 0.9 | 32 | 196 | 95 | 276 |
| [4]N. Obhor | 6/7/2015 | Surface | - | 707,100 | 1.262 | 40 | 194 | 85 | 466 |
| [4]N. Obhor | 6/17/2015 | Surface | - | - | 1.034 | 42 | 185 | 99 | 231 |
| [4]N. Obhor | 7/1/2015 | Surface | 108,740 | 282,450 | 1.162 | 49 | 192 | 105 | 313 |
| [4]N. Obhor | 7/12/2015 | Surface | 87,615 | 252,233 | 1.036 | 50 | 188 | 105 | 477 |
| [4]N. Obhor | 8/3/2015 | Surface | 135,603 | 908,100 | 1.104 | 80 | 209 | 122 | 269 |
| [4]N. Obhor | 8/16/2015 | Surface | 49,770 | 1,764,850 | 1.118 | 71 | 184 | 111 | 284 |
| [4]Saudia | 6/7/2015 | Surface | - | 317,567 | 1.055 | 29 | 172 | 100 | 369 |
| [4]Saudia | 6/17/2015 | Surface | - | - | 1.233 | 46 | 189 | 84 | 183 |
| [4]Saudia | 7/1/2015 | Surface | 61,925 | 583,400 | 1.287 | 44 | 190 | 93 | 152 |
| [4]Saudia | 7/12/2015 | Surface | 137,363 | 1,070,400 | 1.294 | 40 | 159 | 82 | 188 |
| [4]Saudia | 8/3/2015 | Surface | 53,810 | 1,736,450 | 1.164 | 93 | 180 | 111 | 238 |
| [4]Saudia | 8/16/2015 | Surface | 43,060 | 2,182,550 | 1.181 | 83 | 208 | 103 | 276 |
| [5]N. Ohbor | 2/4/2015 | Surface | 91,870 | 1,356,600 | 1.10 | 55 | 214 | 98 | 387 |
| [6]KAUST SW | 5/3/2014 | Surface | 4,766 | 273,400 | 1.42 | 29 | 217 | 119 | 315 |
| [6]KAUST SW | 5/22/2014 | Surface | 9,350 | 236,000 | 1.037 | 55 | 197 | 121 | 252 |
| [6]KAUST SW | 6/11/2014 | Surface | 3,140 | 287,850 | 0.992 | 36 | 246 | 81 | 319 |
| [6]KAUST SW | 7/3/2014 | Surface | 4,958 | 324,600 | 1.085 | 43 | 212 | 151 | 227 |
| [6]KAUST SW | 7/19/2014 | Surface | 11,080 | 389,450 | 0.97 | 53 | 212 | 91 | 233 |



| [6]KAUST SW | 8/18/2014 | Surface | 6,057 | 316,450 | 1.112 | 40 | 201 | 88 | 225 |
|---|---|---|---|---|---|---|---|---|---|
| [6]KAUST SW | 9/18/2014 | Surface | 52,453 | 321,250 | 0.923 | 35 | 193 | 93 | 171 |
| [6]KAUST SW | 10/21/2014 | Surface | 12,228 | 630,600 | 0.831 | 39 | 193 | 108 | 256 |
| [6]KAUST SW | 12/3/2-14 | Surface | 10,673 | 347,133 | 1.004 | 33 | 189 | 101 | 288 |
| [6]KAUST SW | 2/11/2015 | Surface | 12,890 | 292,500 | 1.275 | 36 | 200 | 102 | 343 |
| [6]KAUST SW | 5/21/2015 | Surface | 28,009 | 450,800 | 0.93 | 31 | 177 | 93 | 236 |
| [6]KAUST SW | 8/6/2015 | Surface | 44,153 | 336,900 | 1.041 | 42 | 184 | 86 | 229 |
| [6]KAUST SW | 9/17/2015 | Surface | 52,453 | 297,867 | 1.084 | 28 | 188 | 86 | 230 |
| [7]KAUST SW | 9/30/2016 | Surface | 11,955 | 369,300 | 1.073 | 112 | 429 | 213 | 385 |
| [7]KAUST SW | 10/2/2014 | Surface | 10,600 | 367,000 | 0.993 | 105 | 363 | 193 | 353 |
| [7]KAUST SW | 10/9/2014 | Surface | 17,777 | 368,463 | 0.944 | 140 | 373 | 218 | 335 |
| [7]KAUST SW | 10/16/2014 | Surface | 22,030 | 319,950 | 0.961 | 88 | 340 | 216 | 346 |
| [7]KAUST SW | 10/27/2014 | Surface | 42,550 | 297,700 | 0.917 | 164 | 348 | 260 | 468 |
| [7]KAUST SW | 11/6/2014 | Surface | 86,033 | 587,200 | 0.864 | 73 | 442 | 93 | 470 |
| [7]KAUST SW | 11/17/2014 | Surface | 107,030 | 673,700 | 0.897 | 71 | 374 | 221 | 352 |
| No. Samples | | | 38 | 40 | 42 | 42 | 42 | 42 | 42 |
| Range in values | | | 1,677-137,363 | 112,790-2,182,550 | 0.830-1.420 | 28-164 | 159-442 | 81-260 | 16-477 |
| Average | | | 44,383 | 525,820 | 1 | 62 | 248 | 118 | 271 |

[1] R. Rachman et al. 2015; [2] Dehwah et al. 2015; [3]Dehwah and Missimer 2016, [4]Alsahri et al. 2017,
[5]Dehwah et al 2017; [6]Dehwah and Missimer 2017; [7]Dehwah and Missimer 2015d
Table 2. Regression analysis of selected organic parameters at the 0.05 significance level
0.05 level

| Organic Parameters | Location | $R^2$ | p-value | Significant (?) |
|---|---|---|---|---|
| p-TEP v. Bacteria | Site A | 0.6677 | 0.0039 | Y |
| | Site B | 0.7295 | 0.001656 | Y |
| | Site C | 0.6691 | 0.00383 | Y |
| | Deep Profile (300 m) | 0.3034 | 0.009661 | Y |
| | Nearshore | 0.0593 | 0.158757 | N |
| p-TEP v. Algae | Site A | 0.1011 | 0.37063 | N |
| | Site B | 0.5363 | 0.016017 | Y |
| | Site C | 0.2463 | 0.144607 | N |
| | Deep Profile (300 m) | 0.1495 | 0.083384 | N |
| | Nearshore | 0.0169 | 0.471436 | N |
| c-TEP v. Bacteria | Site A | 0.6677 | 0.0039 | Y |
| | Site B | 0.6430 | 0.005265 | Y |
| | Site C | 0.2474 | 0.143485 | N |
| | Deep Profile (300 m) | 0.5512 | 0.000116 | N |
| | Nearshore | 0.2622 | 0.006329 | N |
| c-TEP v. Algae | Site A | 0.1011 | 0.37063 | N |
| | Site B | 0.2900 | 0.108267 | N |



| | | | | |
|---|---|---|---|---|
| | Site C | 0.0141 | 0.743804 | N |
| | Deep Profile (300 m) | 0.5713 | 7.4E-05 | Y |
| | Nearshore | 0.1476 | 0.057986 | N |
| Biopolymers v. Bacteria | Site A | 0.8166 | 0.000335 | Y |
| | Site B | 0.6726 | 0.003663 | Y |
| | Site C | 0.6868 | 0.003043 | Y |
| | Deep Profile (300 m) | 0.7814 | 1.08E-07 | Y |
| | Nearshore | 0.0123 | 0.495799 | N |
| Biopolymers v. Algae | Site A | 0.5801 | 0.010465273 | Y |
| | Site B | 0.2918 | 0.10701 | N |
| | Site C | 0.2996 | 0.101512 | N |
| | Deep Profile (300 m) | 0.7078 | 1.77E-06 | Y |
| | Nearshore | 0.0107 | 0.537011 | N |
| Biopolymers v. p-TEP | Site A | 0.4890 | 0.024407 | Y |
| | Site B | 0.4824 | 0.0258132 | Y |
| | Site C | 0.4020 | 0.049006 | Y |
| | Deep Profile (300 m) | 0.1551 | 0.077318 | N |
| | Nearshore | 0.0808 | 0.09790 | N |
| Biopolymers v. c-TEP | Site A | 0.4890 | 0.024407 | Y |
| | Site B | 0.3696 | 0.062253 | N |
| | Site C | 0.2590 | 0.13302 | N |
| | Deep Profile (300 m) | 0.5883 | 4.97E-05 | Y |
| | Nearshore | 0.0331 | 0.364097 | N |
| p-TEP v. c-TEP | Site A | 1 | 0 | Y |
| | Site B | 0.362578 | 0.065466 | N |
| | Site C | 0.3798 | 0.057758 | N |
| | Deep Profile (300 m) | 0.1660 | 0.066765 | N |
| | Nearshore | 0.0491 | 0.266597 | N |
| p-TEP v. TOC | Site A | 0.6591 | 0.00434 | Y |
| | Site B | 0.2760 | 0.118919 | N |
| | Site C | 0.0979 | 0.378796 | N |
| | Deep Profile (300 m) | 0.3156 | 0.008046 | Y |
| | Nearshore | 0.0284 | 0.332963 | N |
| c-TEP vs. TOC | Site A | 0.6591 | 0.0043396 | Y |
| | Site B | 0.0431 | 0.565154 | N |
| | Site C | 0.0165 | 0.723942 | N |
| | Deep Profile (300 m) | 0.6698 | 5.79E-06 | Y |
| | Nearshore | 0.1995 | 0.019513 | N |
| Bacteria v. TOC | Site A | 0.7717 | 0.000822 | Y |
| | Site B | 0.2994 | 0.101653 | N |
| | Site C | 0.1294 | 0.307187 | N |
| | Deep Profile (300 m) | 0.7812 | 1.08E-07 | Y |
| | Nearshore | 0.1144 | 0.032827 | Y |
| Algae v. TOC | Site A | 0.0928 | 0.3922134 | N |
| | Site B | 0.4907 | 0.024064 | N |
| | Site C | 0.3188 | 0.089015 | N |

| | Deep Profile (300 m) | 0.6220 | 2.16E-05 | Y |
| | Nearshore | 0.0388 | 0.236167 | N |


**Acknowledgments**

Funding for the offshore sample collection was provided by the King Abdullah University of
Science and Technology Coastal and Marine Resources Core Laboratory. Analytical work was
funded by the Water Desalination and Reuse Center, King Abdullah University of Science and
Technology. Support for DMA was provided by the National Science Foundation (Grants OCE-
0850421 OCE-0430724, OCE-0911031, and OCE-1314642) and National Institutes of Health (NIEHS-
1P50-ES021923-01) through the Woods Hole Center for Oceans and Human Health.

**Conflicts of Interest**
None declared

Figure captions






Fig. 1. Map showing the sampling profile locations in the Red Sea





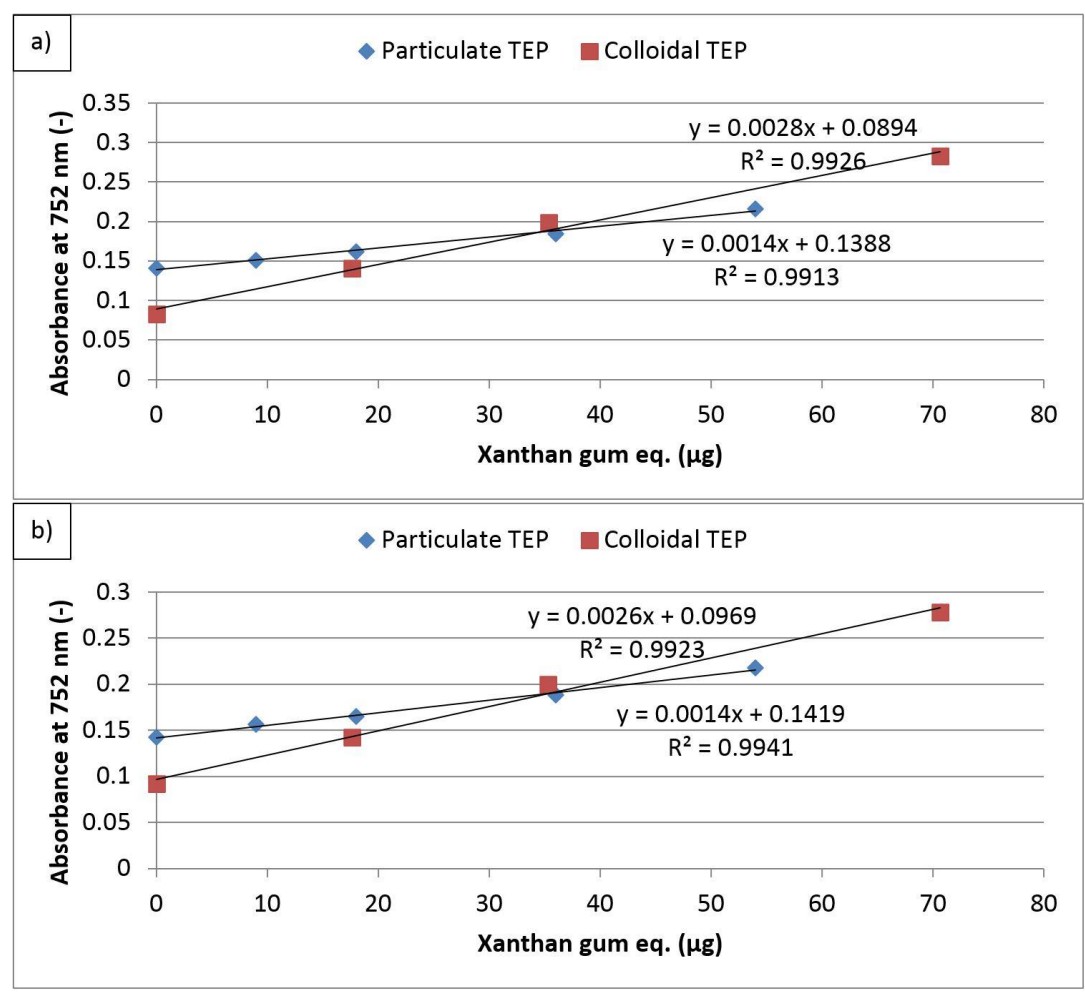


Fig. 2. Xanthan gum standard calibration curves for determination of p-TEP and c-TEP






Fig. 3. Physical data from the three 90 m profiles




Fig. 4. Physical data from the 300 m profile



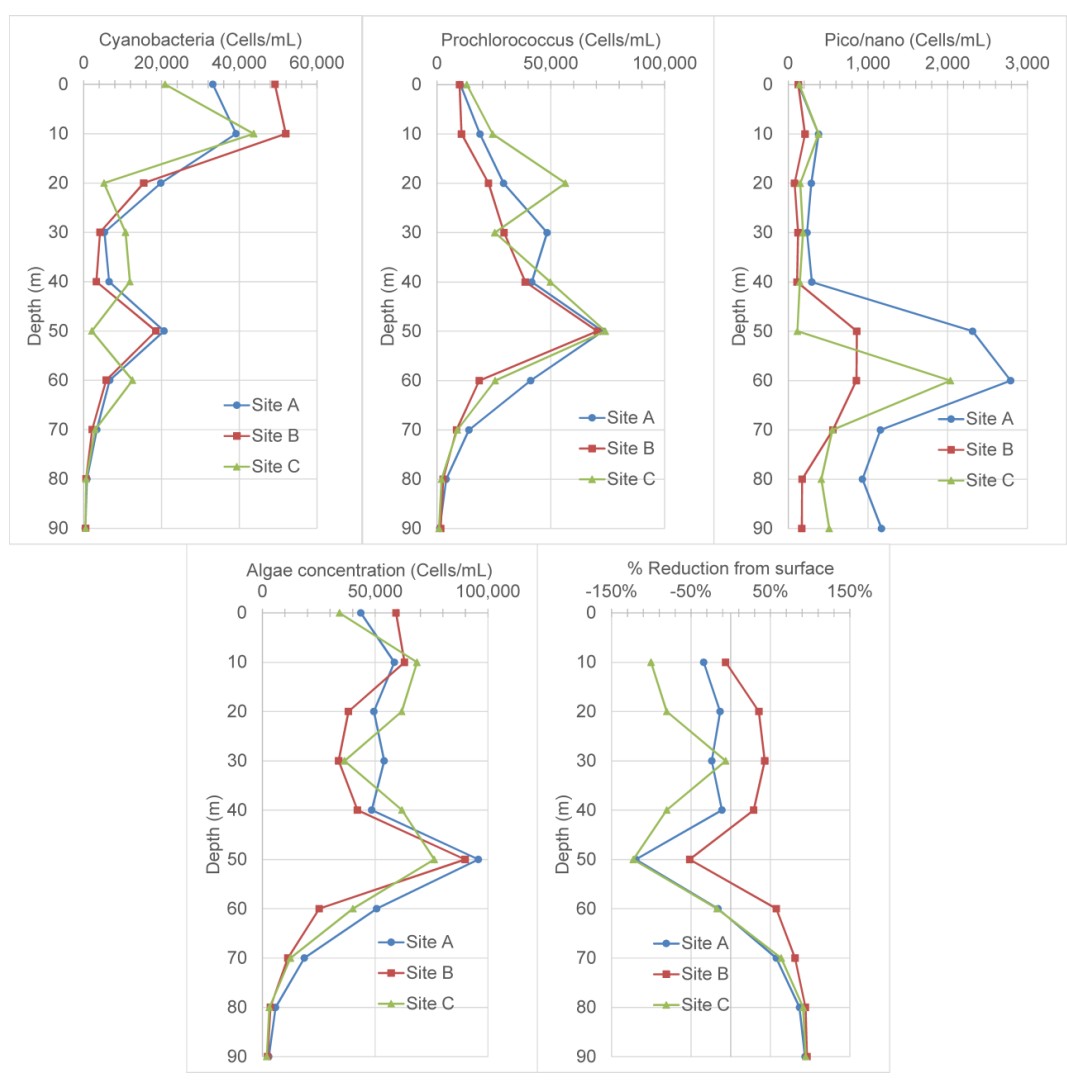


Fig. 5. Algal composition and concentration data from the three 90 m profiles





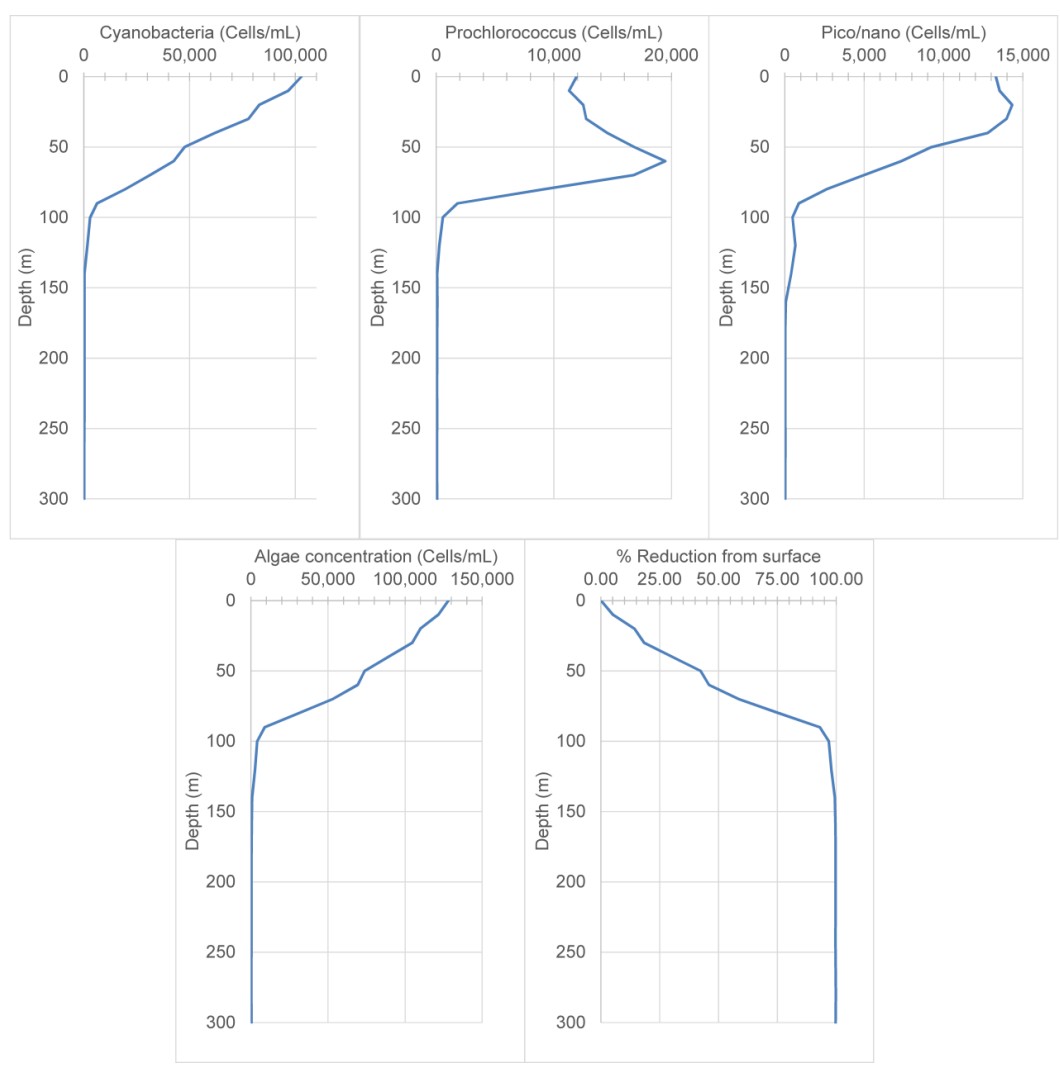


Fig. 6. Algal composition and concentration data from the 300 m profile



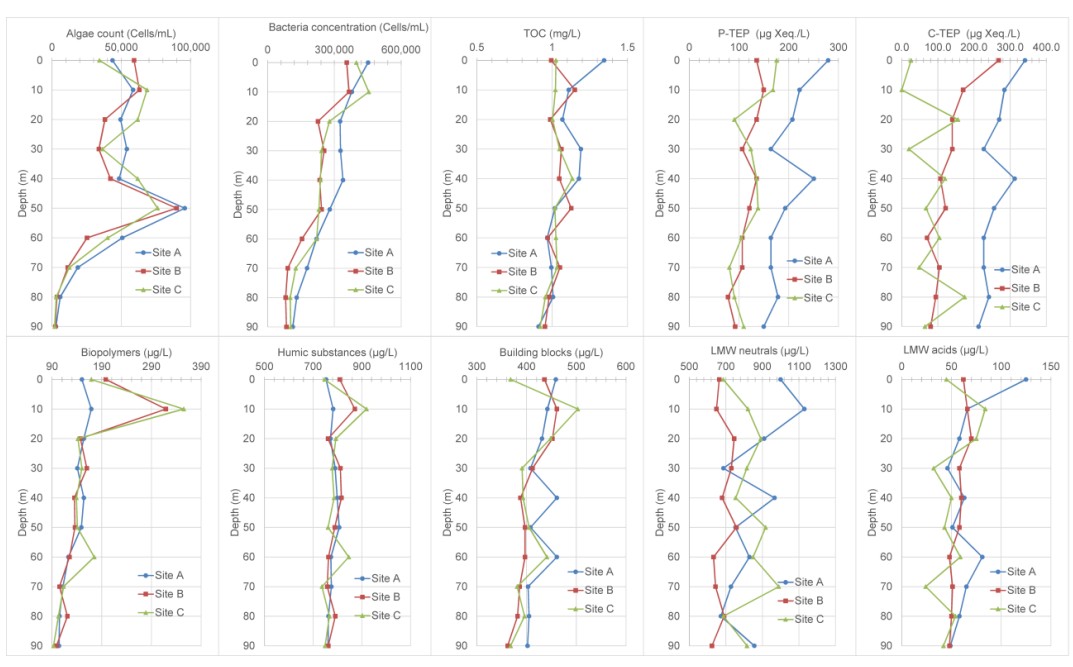


Fig. 7. Organic carbon concentrations for the three 90 m profiles

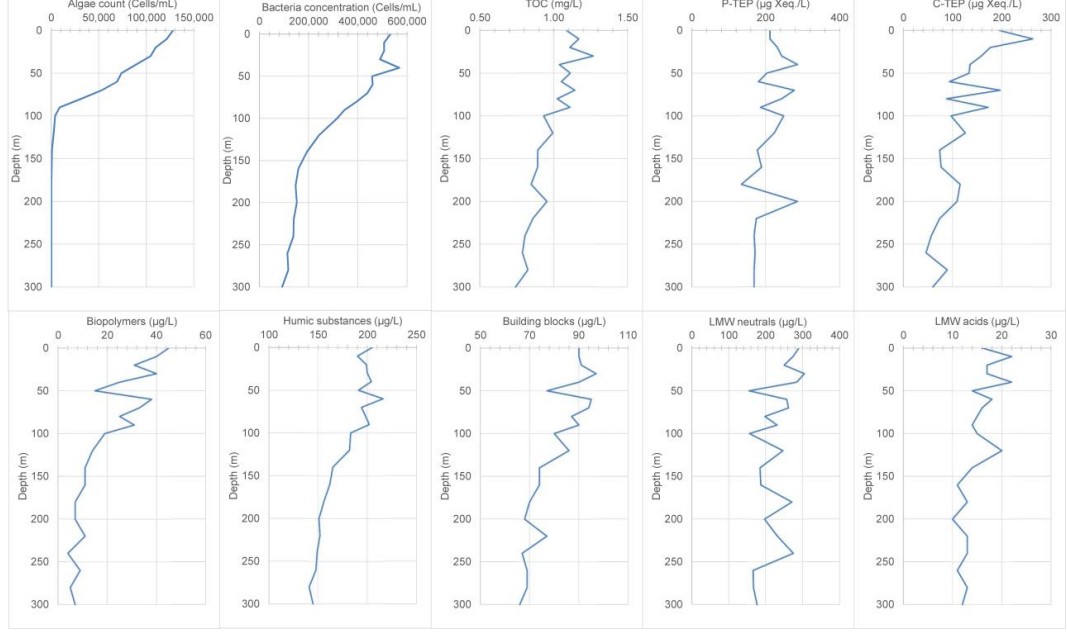


Fig. 8. Organic carbon concentrations from the 300 m profile