# Peer review of "Transparent exopolymer particle binding of organic and inorganic"

_Biogeosciences, 2019_

## Referee Comment (RC1) · Anonymous Referee #1 · 2 Apr 2019

In this manuscript, the authors present survey data on organic matter pools aswell as sensor data from the Red Sea. The study strongly builds upon the circumstance that it is the very first study recording the measured parameters in the given combination at the study region over depth (down to the base of the euphotic zone). The authors are very open in this regard stating themselves that the data presented "have not been collected in a systematic manner with spatial and temporal comparisons to assess the biogeochemical cycles within the Red Sea comprehensively." Hence, the study does not come up with substantial new concepts, but rather provides a starting point for

following up analyses of matter cycling over depth in the study region.

Below are detailed comments. Major issues are that the section describing the statistics needs to be elaborated to achieve appropriate quality of the paper. The study design is rather unbalanced, with three shallower stations sampled during spring and one deep station sampled during winter. Additional data from unrelated surveys in the region were compiled and included in the manuscript as an additional data source for measurements at surface depths.

*******************General remarks*******************

- The title is rather misleading since the binding capacity of TEP was not examined explicitly, as the title implies. In addition the term 'inorganic particles' seems not to be suitable here as solely the organic fraction is analysed.

- The description of the results is not completely balanced. Some parts should contain fewer details while other more relevant parts are only scratched at the surface (see further down for detailed comment on this).

- The authors should be more careful with the use of literature. For example references indicated in the text are missing in the reference list (check for example Villacorte et al. 2009).

- Overall the manuscript is clearly structured. The text is written in a honest way and the results are discussed critically.

*******************Specific remarks*******************

- L48: Please specify the difference between sediments and POM. In my opinion the term sediments is not ideal for matter within the water column, especially not if it is contrasted to POM. The authors should consider to choose a different term.

- L55 'some' organic matter: Please be more precise.

- Fig.1: Where is site D? The authors should be consistent and add similar labelling as

for station A-C. In general the labelling of sites does not seem optimal. My suggestion is to use different colors for the sites sampled during the present study in contrast to the stations incorporated from previous studies.

- Fig.2: This figure should rather go into the supplementary material, as it is only a technical validation/quality assessment rather than adding to the results of the study.

- L223 Overall remark on Section 2.6-Statistical methods: This section is poorly described. How did you deduce statistical significance from a scatter plot? Besides a p-value also the statistical method applied has to be indicated.

- L229 Overall remark on Section 3- Results: A substantial amount of text in the results is spent on the description of the physical/sensor data. However, these data do not contribute substantially to the following discussion. I would suggest to shorten the respective results section and to remove 'hitchhiking' parameters such as pH, dissolved oxygen and turbidity from Fig.2 and Fig.3. I guess these parameters have been recorded and published for the study region before. Otherwise the authors should put more effort into putting these parameters into context in their discussion. From the TS profiles it looks like different watermasses could be present, which could have implications for organic matter cycling. In terms of statistics the study is scratching only at the surface. My suggestion is to include further statistical tests evaluating the difference between shallow and deeper water layers. One option would be for example to pool the three stations A-C for a comparison of the different OM parameters at minimal depth against maximal depths.

- L232 biospherical licor: This aspect could be skiped from my perspective (see comment above).

- L244 'a slightly lower salinity gradient': What is the precision of the measurement? I guess the observed variation lies within the methodological range. Please correct me if I am wrong.

- L254 oxygen variability: Was the CTD device 'acclimatised' in the water for several minutes before starting to run the profile in order to avoid methodological biases?

- Fig.8: The figure headers are quite hard too read. The authors should consider to increase the font size.

- Overall remark on Figures: The figure captions should be more elaborated in order to transport a message. Identical units should be used for all parameters within the same figures if applicable. For example in Fig.7 and Fig.8 $\mu$g/L should be used also for TOC (instead of mg/L) to facilitate comparability across parameters.

- L368 'highly effective': How do you define high effectiveness here?

- L410: As you state that TEP is presumably a significant part of TOC, it would be valuable to also calculate the respective fractions and indicate them in the manuscript (maybe even as an additional figure).

- L414: Can methodological issues such as a bursted filter be excluded?

- L464: I would be careful here as this aspect was not measured within the study scope. The authors should replace 'which are food' with 'which can be food'.

- L491 'unusual result': Nutrient concentrations would be interesting to check in this regard. The chlorophyll maximum seems to lay quite deep at station A-C.

- LL503-504 offshore vs. nearshore: This statement in the conclusion can be misleading, as season may be the more influential factor than offshore versus nearshore.

- L505: Which irregularities?

*******************Technical corrections*******************

- L57 LANGUAGE: The term 'caused' seems not fitting here, probably better to be replaced with e.g. 'formed'.

- L116 TYPO: 'the characterize'

- L156 TYPO: 'there different'

---

## Referee Comment (RC2) · Anonymous Referee #2 · 5 Jul 2019

**General comments:**

Dehwah et al. investigated the vertical profiles of concentrations of transparent exopolymer particles (TEP) and other forms of organic matter in the Saudi coast of the Red Sea as well as microbial cell abundances and several kinds of hydrological parameters. The results showed that particulate TEP concentration (p-TEP) decreased at a diminished rate with depth compared to other organic matter compositions, indicating that it persists in moving organic carbon deeper into the water column. The vertical distributions of diverse organic matter compositions, the relationships between TEP and microbes, and the microbial impacts on organic carbon transport into the ocean are indeed worth studying. However, I am not convinced for its publication in BG with the current form. My major concerns are shown below.

1. Many paragraphs need to be re-written for a clear logic. The results section need to be re-organized to focus on main/important findings.

2. The uses of algae, picoplankton, cyanobacteria, and *Prochlorococcus* are completely confusing, making this manuscript hard to understand.

3. The analysis is too simple and the conclusions are too speculative only based on regression analysis.

**Specific comments:**

Lines 54 to 56: What's the meaning of "biomass"? particulate organic material or bacteria? Hard to understand.

Lines 91 to 94: This paragraph should be incorporated into the next paragraph.

Lines 110 to 113: The compiled data were published already? If yes, citations need to be added. If not, use the data as new data.

Line 121: Where is the site D? not shown in Fig 1.

Lines 136 to 140: This paragraph should be incorporated into the previous sensors part.

Lines 144 to 147: This paragraph should be incorporated into the next paragraph.

Lines 146 to 147: delete.

Lines 148 to 149: This sentence should be placed before "Algal cell counting…..".

Lines 151 to 152: This sentence should be placed in the next paragraph of bacteria counting.

Lines 156 to157: The use of algae, cyanobacteria, *Prochlorococcus*, and pico/nanoplankton is very confusing! Cyanobacteria include *Prochlorococcus* and *Synechococcus*; picoplankton include cyanobacteria and pico-eukaryotes. Algae include cyanobacteria here? These confusions need to be clarified throughout the manuscript.

Lines 191 to 192: The size ranges were repeated in the introduction section. Delete here.

Line 231: This part is too long and wordy. The authors should emphasize the key points related to the conclusions.

Lines 283 and 366: Descriptions of the two sections are very confusing. Please see my comment above.

Line 285: the sum of what?

Line 286: Cyanobacterial abundances are not shown in Table 1.

Lines 348 and 389 to392: Which samples are offshore samples and which are nearshore samples? clarify.

Line 364: Discussion section included many results descriptions and needs to be reorganized to focus on the deep discussion.

Lines 396 to 401, 405 to 405, 437 to 439, and 454 to 458: too speculative! There is no supporting evidence. Only vertical distributions of parameters and regression analysis are not enough.

Lines 406 to 408: Is there overlap between biopolymers and TEP? If so, there is an internal correlation between them. The significance of regression analysis means nothing.

Lines 431 to 432: The data are from the published paper? insert citations.

Lines 464 to 471: this part belongs to the result descriptions.

Table 1: give references; indicate what mean for total algae.

Table 2: Which kind of regression analysis? Which samples are nearshore and offshore samples respectively? Why do many p-values of <0.05 correspond to "N" (significant)?

Fig. 1: Site D is now shown.

Fig. 2: what do a) and b) indicate?

Fig. 4: the unit of salinity should be ppt.

Figs 5 and 6: need error bars. The figure legend is unclear!

Fig. 6: use dot-lines.

Fig8: needs error bars. Algae count figure is repeated.

---

## Author Comment (AC1) · 25 Jul 2019

Reviewer #2. Comments and Responses General comments: Dehwah et al. investigated the vertical profiles of concentrations of transparent exopolymer particles (TEP) and other forms of organic matter in the Saudi coast of the Red Sea as well as microbial cell abundances and several kinds of hydrological parameters. The results showed that particulate TEP concentration (p-TEP) decreased at a diminished rate with depth compared to other organic matter compositions, indicating that it persists in moving organic carbon deeper into the water column. The vertical distributions of

diverse organic matter compositions, the relationships between TEP and microbes, and the microbial impacts on organic carbon transport into the ocean are indeed worth studying. However, I am not convinced for its publication in BG with the current form. My major concerns are shown below. Response: We agree with the description of the paper content. We will address the issues the reviewer has raised. 1. Many paragraphs need to be re-written for a clear logic. The results section needs to be re-organized to focus on main/important findings. Response: We will try to do so, but are unsure on what reorganization the reviewer is suggesting. 2. The uses of algae, picoplankton, cyanobacteria, and Prochlorococcus are completely confusing, making this manuscript hard to understand. Response: We do not agree. Another reviewer has not raised this issue and other members of our faculty of marine science believe that it is quite clear. 3. The analysis is too simple and the conclusions are too speculative only based on regression analysis. Response: It is most difficult to respond to this comment in that the reviewer makes no suggestions regarding what statistical approaches we should use. We have added some additional statistical analysis including multi-variant analysis and ANOVA. The results have confirmed what we found in the various regression analyses made. We have added the spreadsheet with the detailed analyses to the Supplemental Information File. Specific comments: Lines 54 to 56: What's the meaning of "biomass"? particulate organic material or bacteria? Hard to understand. Response: We do not understand what the reviewer is suggesting. The sentence appears quite clear to us. Lines 91 to 94: This paragraph should be incorporated into the next paragraph. Response: We added the single sentence paragraph to the beginning of the next paragraph. Lines 110 to 113: The compiled data were published already? If yes, citations need to be added. If not, use the data as new data. Response: The references have been added. Line 121: Where is the site D? not shown in Fig 1. Response: The location of site D has been added to Figure 1. Lines 136 to 140: This paragraph should be incorporated into the previous sensors part. Response: We do not agree. The paragraph must stand along so that the English is correct. Long paragraphs with multiple subjects are not desirable.

Lines 144 to 147: This paragraph should be incorporated into the next paragraph. Response: We have merged the paragraphs as suggested. Lines 146 to 147: delete. Response: We edited the paragraph to remove the redundancy. Lines 148 to 149: This sentence should be placed before "Algal cell counting……". Response: Additional editing was done on the paragraph. Lines 151 to 152: This sentence should be placed in the next paragraph of bacteria counting. Response: Additional editing was done on the paragraph. Lines 156 to157: The use of algae, cyanobacteria, Prochlorococcus, and pico/nanoplankton is very confusing! Cyanobacteria include Prochlorococcus and Synechococcus; picoplankton include cyanobacteria and pico-eukaryotes. Algae include cyanobacteria here? These confusions need to be clarified throughout the manuscript. Response: We restructured the sentence to note that cyanobacteria are not algae. Lines 191 to 192: The size ranges were repeated in the introduction section. Delete here. Response: The sentence was repeated. Line 231: This part is too long and wordy. The authors should emphasize the key points related to the conclusions. Response: We do not agree. The change in the slope at 115 m shows the possible presence of two water masses which requires us to carefully describe the profile conditions. We added minor text at the end of the halocline description. Lines 283 and 366: Descriptions of the two sections are very confusing. Please see my comment above. Response: We added site D to the text, otherwise the text is clear. Line 285: the sum of what? Response: As stated, the sum of the total algae and cyanobacteria. Line 286: Cyanobacterial abundances are not shown in Table 1. Response: We corrected the heading to show it as Total Algae and Cyanobacteria in Table 1. Lines 348 and 389 to 392: Which samples are offshore samples and which are nearshore samples? clarify. Response: We clarified by adding a reference to Table 1 in the first sentence of the paragraph. Line 364: Discussion section included many results descriptions and needs to be reorganized to focus on the deep discussion. Response: We do not agree that all of the text should be on the deep water profile. We are not sure why the reviewer is suggesting these changes which are not defined. Lines 396 to 401, 405 to 405, 437 to 439, and 454 to 458: too speculative! There is no

supporting evidence. Only vertical distributions of parameters and regression analysis are not enough. Response: We have clearly stated the evidence for our interpretation of the data. What additional evidence could be collected? Perhaps the reviewer could provide us with some alternative interpretations of the data that we could add to the text. Additional statistical work on the dataset would not yield any significant results due to the small sample size in the site D profile. Lines 406 to 408: Is there overlap between biopolymers and TEP? If so, there is an internal correlation between them. The significance of regression analysis means nothing. Response: TEP consists of larger acidic polysaccharides and proteins as well as smaller molecules. Some of the TEP material occurs with the biopolymer fraction of NOM and many proteins are likely contained with the humic substances. The reason that there is a statistically significant relationship between p-TEP and the biopolymer fraction is the common occurrence of the polysaccharides. The statistical relationship is significant. Lines 431 to 432: The data are from the published paper? insert citations. Response: The references have been added. Lines 464 to 471: this part belongs to the result descriptions. Response: We do not agree. Table 1: give references; indicate what mean for total algae. Response: The references are given at the end of the table and correlated to the numbers. Table 2: Which kind of regression analysis? Which samples are nearshore and offshore samples respectively? Why do many p-values of <0.05 correspond to "N" (significant)? Response: The regression tests the relationship between the two parameters. The p-value used for determine of significance is a standard. Fig. 1: Site D is now shown. Response: This has been corrected. Fig. 2: what do a) and b) indicate? Response: The curve labelled a is for the analyses conducted at samples from sites A, B, and C. The b label was developed for site D. We have added text to the table to cover this issue. The a) and b) indicate the calibration curves used for the study. The reason for two calibration curves was: the alcian blue solution prepared for staining should be used for no more than 1 month, otherwise the alcian blue solution should be filtered with 0.2 $\mu$m filters and a new calibration curve should be prepared. In this study, the execution period was more than 1 month, so two

calibration curves were prepared. Fig. 4: the unit of salinity should be ppt. Response: The PPM has been corrected to ppt. Figs 5 and 6: need error bars. The figure legend is unclear! Response: Error bars are not used on flow cytometer data by convention. As a group, we have published 15 papers with this type of data and no journal has ever required error bars. Fig. 6: use dot-lines. Response: No. We do not agree. Fig8: needs error bars. Algae count figure is repeated. Response: Error bars are not used on flow cytometer data by convention. As a group, we have published 15 papers with this type of data and no journal has ever required error bars, particularly for LCOCD data. The algal data was presented in this figure a second time for comparison.

Please also note the supplement to this comment:
https://www.biogeosciences-discuss.net/bg-2019-59/bg-2019-59-AC1-supplement.pdf

[Figure]

**Supplement:**

**Statistical relationship between the variables**

| | Depth (m) | P-TEP (µg X | C-TEP (µg Xeq./L)-Site A | Bacterial count (cells/ml)-Site A | Total algae (events/ml)-Site A | TOC (mg/L | Biopolymers | |
|---|---|---|---|---|---|---|---|---|
| Site **A** | 0 | 279.36 | 340.9 | 451,650 | 43,612 | 1.345 | 150 | **A-C** |
| | 10 | 221.76 | 283.8 | 378,300 | 58,487 | 1.11 | 169 | |
| | 20 | 207.36 | 269.6 | 325,700 | 49,427 | 1.068 | 154 | Dependent |
| | 30 | 164.16 | 226.8 | 327,433 | 53,980 | 1.191 | 141 | |
| | 40 | 250.56 | 312.4 | 338,700 | 48,440 | 1.178 | 154 | |
| | 50 | 192.96 | 255.3 | 279,000 | 95,772 | 1.018 | 149 | |
| | 60 | 164.16 | 226.8 | 220,000 | 50,663 | 0.967 | 123 | |
| | 70 | 164.16 | 226.8 | 177,500 | 18,593 | 0.995 | 112 | |
| | 80 | 178.56 | 241.0 | 130,400 | 5,853 | 1.006 | 105 | |
| | 90 | 149.76 | 212.5 | 113,233 | 2,790 | 0.91 | 104 | |
| Site **B** | 0 | 135.36 | 268.1 | 355,567 | 59,230 | 0.994 | 198 | |
| | 10 | 149.76 | 169.7 | 365,900 | 63,002 | 1.15 | 319 | |
| | 20 | 135.36 | 139.8 | 226,300 | 38,177 | 0.9875 | 147 | |
| | 30 | 106.56 | 139.8 | 254,233 | 33,817 | 1.0605 | 160 | **A-D** |
| | 40 | 135.36 | 107.0 | 234,500 | 42,150 | 1.047 | 135 | |
| | 50 | 120.96 | 121.2 | 242,600 | 89,885 | 1.127 | 136 | |
| | 60 | 106.56 | 69.9 | 154,467 | 25,218 | 0.97 | 125 | |
| | 70 | 106.56 | 104.1 | 90,733 | 11,297 | 1.051 | 105 | |
| | 80 | 77.76 | 94.1 | 80,867 | 3,490 | 0.98 | 121 | |
| | 90 | 92.16 | 79.9 | 84,767 | 2,285 | 0.9525 | 99 | |
| Site **C** | 0 | 175.68 | 25.7 | 398,700 | 34,210 | 1.025 | 169 | |
| | 10 | 168.48 | 0.0 | 454,550 | 68,550 | 1.022 | 355 | |
| | 20 | 90.72 | 155.5 | 277,950 | 61,795 | 1.002 | 142 | |
| | 30 | 123.84 | 20.0 | 242,200 | 36,368 | 1.049 | 150 | |
| | 40 | 135.36 | 119.8 | 237,233 | 61,820 | 1.134 | 139 | **A-E** |
| | 50 | 138.24 | 67.0 | 230,700 | 76,185 | 1.018 | 141 | |
| | 60 | 105.12 | 104.1 | 220,333 | 40,115 | 1.025 | 175 | |
| | 70 | 80.64 | 48.5 | 126,433 | 12,430 | 1.032 | 113 | |
| | 80 | 90.72 | 174.0 | 101,500 | 3,075 | 0.955 | 103 | |
| | 90 | 109.44 | 64.2 | 103,867 | 1,900 | 0.921 | 93 | |
| | 0 | 211.71 | 193.8 | 532500 | 128077 | 1.09 | 45 | |
| | 10 | 211.71 | 263.1 | 507100 | 121507 | 1.17 | 40 | |
| | 20 | 231.71 | 176.9 | 506750 | 109840 | 1.11 | 31 | |
| | 30 | 243.14 | 158.5 | 490050 | 104593 | 1.27 | 40 | |
| | 40 | 286.00 | 135.4 | 568500 | 89153 | 1.04 | 25 | |
| | 50 | 203.14 | 133.8 | 458000 | 73797 | 1.11 | 15 | |
| | 60 | 180.29 | 93.8 | 460100 | 69253 | 1.05 | 38 | |
| | 70 | 277.43 | 196.9 | 438800 | 52973 | 1.14 | 33 | |
| | 80 | 243.14 | 87.7 | 395650 | 31363 | 1.02 | 25 | |
| Deep | 90 | 186.00 | 172.3 | 345750 | 8907 | 1.11 | 31 | |

| | | | | | | |
|---|---|---|---|---|---|---|
| **Deep Profile** | 100 | 248.86 | 96.9 | 315450 | 4067 | 0.93 | 19 |
| | 120 | 223.14 | 126.2 | 242300 | 2642 | 0.99 | 14 |
| | 140 | 177.43 | 73.8 | 192700 | 832 | 0.89 | 11 |
| | 160 | 188.86 | 76.9 | 157850 | 467 | 0.89 | 11 |
| | 180 | 134.57 | 115.4 | 146967 | 315 | 0.85 | 7 |
| | 200 | 286.00 | 109.2 | 152033 | 332 | 0.95 | 7 |
| | 220 | 174.57 | 73.8 | 139067 | 303 | 0.86 | 11 |
| | 240 | 168.86 | 56.9 | 138250 | 342 | 0.81 | 4 |
| | 260 | 171.71 | 46.2 | 114050 | 257 | 0.79 | 9 |
| | 280 | 168.86 | 89.2 | 117000 | 240 | 0.83 | 5 |
| | 300 | 168.86 | 60.0 | 92350 | 277 | 0.74 | 7 |
| **Near Shore (Surface)** | | 121 | 73 | 196,377 | 3,603 | 0.94 | 90 |
| | | 157 | 122 | 264,728 | 1,677 | 1.02 | 116 |
| | | 123 | 130 | 179,837 | 14,956 | 0.88 | 63 |
| | | 53 | 56 | 317,174 | 23,773 | 0.83 | 84 |
| | | 318 | 90 | 520,350 | 129,738 | 1.1 | 57 |
| | | 249 | 120 | 254,450 | 89,033 | 1 | 44 |
| | | 255 | 115 | 216,400 | 42,923 | 0.9 | 32 |
| | | 142 | 189 | 282,450 | 108,740 | 1.162 | 49 |
| | | 146 | 213 | 583,400 | 61,925 | 1.287 | 44 |
| | | 347 | 287 | 1,736,450 | 53,810 | 1.164 | 93 |
| | | 0 | 0 | 2,182,550 | 43,060 | 1.181 | 83 |
| | | 261 | 132 | 1,356,600 | 91,870 | 1.1 | 55 |
| | | 278 | 100 | 273,400 | 4,766 | 1.42 | 29 |
| | | 346 | 97 | 236,000 | 9,350 | 1.037 | 55 |
| | | 229 | 170 | 287,850 | 3,140 | 0.992 | 36 |
| | | 85 | 127 | 324,600 | 4,958 | 1.085 | 43 |
| | | 99 | 117 | 389,450 | 11,080 | 0.97 | 53 |
| | | 82 | 112 | 316450 | 6,057 | 1.112 | 40 |
| | | 97 | 69 | 321,250 | 52,453 | 0.923 | 35 |
| | | 213 | 50 | 630,600 | 12,228 | 0.831 | 39 |
| | | 138 | 43 | 347,133 | 10,673 | 1.004 | 33 |
| | | 182 | 87 | 292,500 | 12,890 | 1.275 | 36 |
| | | 143 | 50 | 450,800 | 28,009 | 0.93 | 31 |
| | | 186 | 36 | 336,900 | 44,153 | 1.041 | 42 |
| | | 300 | 105 | 297,867 | 52,453 | 1.084 | 28 |

| | P-TEP (µg Xeq. | C-TEP (µg Xeq. | Bacterial coun | Total algae (e | TOC (mg/L) | Biopolymers |
|---|---|---|---|---|---|---|
| P-TEP (µg Xeq./L | 1.000 | | | | | |
| C-TEP (µg Xeq./L | 0.689 | 1.000 | | | | |
| Bacterial count ( | 0.679 | 0.304 | 1.000 | | | |
| Total algae (ever | 0.362 | 0.185 | 0.690 | 1.000 | | |
| TOC (mg/L) | 0.612 | 0.417 | 0.635 | 0.437 | 1.000 | |
| Biopolymers | 0.228 | -0.100 | 0.707 | 0.509 | 0.272 | 1.000 |

Absolute value of 0-0.19 is regarded as very weak, 0.2-0.39 as weak, 0.6-0.79 as strong and 0.8-1 as very
But these are rather arbitrary limits, and the context of the results should be considered.

| | P-TEP (µg Xeq. | C-TEP (µg Xeq. | Bacterial coun | Total algae (e | TOC (mg/L) | Biopolymers |
|---|---|---|---|---|---|---|
| P-TEP (µg Xeq./L | 1.000 | | | | | |
| C-TEP (µg Xeq./L | 0.373 | 1.000 | | | | |
| Bacterial count ( | 0.623 | 0.337 | 1.000 | | | |
| Total algae (ever | 0.280 | 0.354 | 0.792 | 1.000 | | |
| TOC (mg/L) | 0.324 | 0.529 | 0.668 | 0.646 | 1.000 | |
| Biopolymers | -0.363 | 0.143 | 0.081 | 0.235 | 0.337 | 1.000 |

| | P-TEP (µg Xeq | C-TEP (µg Xeq | Bacterial coun | Total algae (e | TOC (mg/L) | Biopolymers |
|---|---|---|---|---|---|---|
| P-TEP (µg Xeq./L)-Site A | | | | | | |
| C-TEP (µg Xeq./L | 0.327 | | | | | |
| Bacterial count ( | 0.152 | 0.075 | | | | |
| Total algae (ever | 0.320 | -0.019 | 0.016 | | | |
| TOC (mg/L) | 0.259 | 0.422 | 0.374 | -0.169 | | |
| Biopolymers | -0.263 | 0.207 | -0.006 | -0.304 | 0.209 | |

All<0.8          No Multicollinearity

| A- D (upto100m) | P-TEP (µg Xeq./L) | (µg Xeq./L) | Bacterial cue (events/r | TOC (mg/L) | Biopolymers |
|---|---|---|---|---|---|
| P-TEP (µg Xeq./L)-Site | 1.000 | | | | |
| C-TEP (µg Xeq./L)-Site | 0.461 | 1.000 | | | |
| Bacterial count (cells/ | 0.778 | 0.229 | 1.000 | | |
| Total algae (events/m | 0.478 | 0.045 | 0.503 | 1.000 | |
| TOC (mg/L) | 0.517 | 0.426 | 0.588 | 0.133 | 1.000 | |
| Biopolymers | -0.365 | -0.052 | -0.179 | -0.236 | -0.015 | 1.000 |

strong correlation,

---

## Author Comment (AC2) · 25 Jul 2019

A. Dehwah et al. Anonymous Referee #1 In this manuscript, the authors present survey data on organic matter pools as well as sensor data from the Red Sea. The study strongly builds upon the circumstance that it is the very first study recording the measured parameters in the given combination at the study region over depth (down to the base of the euphotic zone). The authors are very open in this regard stating themselves the data presented "have not been collected in a systematic manner with spatial and temporal comparisons to assess the biogeochemical cycles within the Red Sea comprehensively." Hence, the study does not come up with substantial new concepts, but rather provides a starting point for following up analyses of matter cycling over depth in the study region. Below are detailed comments. Major issues are that the section describing the statistics needs to be elaborated to achieve appropriate quality of the paper. The study design is rather unbalanced, with three shallower stations sampled during spring and one deep station sampled during winter. Additional data from unrelated surveys in the region were compiled and included in the manuscript as an additional data source for measurements at surface depths. Response: We agree with this statement of paper content. *******************General remarks******************* - The title is rather misleading since the binding capacity of TEP was not examined explicitly, as the title implies. In addition, the term 'inorganic particles' seems not to be suitable here as solely the organic fraction is analysed. Response: We have changed the title of the paper as suggested by the reviewer. The new title is: Transparent exopolymer particle occurrence and interaction with algae, bacteria, and the fractions of organic carbon: Implications for downward transport of biogenic materials.

- The description of the results is not completely balanced. Some parts should contain fewer details while other more relevant parts are only scratched at the surface (see further down for detailed comment on this). Response: We will address each suggested revision suggested. -The authors should be more careful with the use of literature. For example, references indicated in the text are missing in the reference list (check for example Villacorte et al. 2009). Response: The missing reference has been added. - Overall the manuscript is clearly structured. The text is written in an honest way and the results are discussed critically. Response: We agree. *******************Specific remarks******************* - L48: Please specify the difference between sediments and POM. In my opinion the term sediments is not ideal for matter within the water column, especially not if it is contrasted to POM. The authors should consider to choose a

different term. Response: We have eliminated the term sediment and revised the sentence to read: "... cycling of nutrients and particulate organic matter (POM) in general." - L55 'some' organic matter: Please be more precise. Response: "some" has been deleted. - Fig.1: Where is site D? The authors should be consistent and add similar labelling as for station A-C. In general, the labelling of sites does not seem optimal. My suggestion is to use different colors for the sites sampled during the present study in contrast to the stations incorporated from previous studies. Response: We have added the missing label for site D on figure 1 and made the previous study location a blue color as suggested. - Fig.2: This figure should rather go into the supplementary material, as it is only a technical validation/quality assessment rather than adding to the results of the study. Response: We do not agree that these parameters should be omitted from the text on the paper in that the pH and suspended sediment contribute to binding with TEP and can alter the rate of downward transport. All of the parameter should be shown for completeness. - L223 Overall remark on Section 2.6-Statistical methods: This section is poorly described. How did you deduce statistical significance from a scatter plot? Besides a p-value also the statistical method applied has to be indicated. Response: This section has been revised as suggested and a new, more advanced statistical analysis has been added. - L229 Overall remark on Section 3- Results: A substantial amount of text in the results is spent on the description of the physical/sensor data. However, these data do not contribute substantially to the following discussion. I would suggest to shorten the respective results section and to remove 'hitchhiking' parameters such as pH, dissolved oxygen and turbidity from Fig.2 and Fig.3. I guess these parameters have been recorded and published for the study region before. Otherwise the authors should put more effort into putting these parameters into context in their discussion. From the TS profiles it looks like different water masses could be present, which could have implications for organic matter cycling. In terms of statistics the study is scratching only at the surface. My suggestion is to include further statistical tests evaluating the difference between shallow and deeper water layers. One option would

before example to pool the three stations A-C for a comparison of the different OM parameters at minimal depth against maximal depths. Response: The reviewer may be correct that there appears to be different water masses shown in the site D profile from 0 to 120 m verses 120 to 300 m. However, the location is not far from the coastline and no currents are known to exist in the region to create a second water mass.. - L232 biospherical licor: This aspect could be skipped from my perspective (see comment above). Response: We do not agree and we believe it should remain for completeness. - L244 'a slightly lower salinity gradient': What is the precision of the measurement? I guess the observed variation lies within the methodological range. Please correct me if I am wrong. Response: The precision of the instrument is based on the conductivity, temperature and pressure. The measurement error range for conductivity is $\pm 0.0003$ S/m, for temperature is $\pm 0.001$ oC, for pressure is $\pm$o,o15% of full range. - L254 oxygen variability: Was the CTD device 'acclimatised' in the water for several minutes before starting to run the profile in order to avoid methodological biases? Response: The device was properly used based on the recommended standard practices.While on deck the instrument is always stored in water. - Fig.8: The figure headers are quite hard too read. The authors should consider to increase the font size. - Overall remark on Figures: The figure captions should be more elaborated in order to transport a message. Identical units should be used for all parameters within the same figures if applicable. For example in Fig.7 and Fig.8 $\mu$g/L should be used also for TOC (instead of mg/L) to facilitate comparability across parameters. Response: We have increased the font sizes as recommended added and we changed the TOC values to ug/L. However, the PDF review copy is not to the resolution that would be published in the journal - L368 'highly effective': How do you define high effectiveness here? Response: The language was changed to read . . .in this study was used in . . . - L410: As you state that TEP is presumably a significant part of TOC, it would be valuable to also calculate the respective fractions and indicate them in the manuscript (maybe even as an additional figure). Response: Since the composition of TEP is known to vary in the extreme, this type

of calculation cannot be done to any reasonable degree of accuracy. If the reviewer has some formula to do this calculation, please send it to us to assist us. - L414: Can methodological issues such as a bursted filter be excluded? Response: No, the laboratory analyses were performed very carefully and a failure of a filter would have been noticed. -L464: I would be careful here as this aspect was not measured within the study scope. The authors should replace 'which are food' with 'which can be food'. Response: The language change was made as requested. - L491 'unusual result': Nutrient concentrations would be interesting to check in this regard. The chlorophyll maximum seems to lay quite deep at station A-C. Response: We do not have data to check this. - LL503-504 offshore vs. nearshore: This statement in the conclusion can be misleading, as season may be the more influential factor than offshore versus nearshore. Response: We added a sentence "Seasonal differences during sampling could have also impacted the results" to address this issue. - L505: Which irregularities? Response: We reworded the sentence to respond to this comment as "Differences in local conditions, such as circulation and anthropogenic influences, in the nearshore zone can cause large variations in the organic parameters measured, not allowing statistically-significant relationships to be established. ". ****************Technical corrections**************** - L57 LANGUAGE: The term 'caused' seems not fitting here, probably better to be replaced with e.g. 'formed'. Response: Changed to "formed" - L116 TYPO: 'the characterize' C4 Response: The text was changed to read"... characterization of the natural...". - L156 TYPO: 'there different' Response: Corrected. It now reads "The different types of algae,...".

Please also note the supplement to this comment:
https://www.biogeosciences-discuss.net/bg-2019-59/bg-2019-59-AC2-supplement.pdf

---

## Author Comment (AC3) · 25 Jul 2019

eTTransparent exopolymer particle occurrence and interaction with algae, bacteria, and the fractions of organic carbonbinding of organic and inorganic particles in the Red Sea: Implications for downward transport of biogenic materials

Abdullah H. A. Dehwah1,6, Donald M. Anderson2, Sheng Li1,4, Francis L. Mallon3, Zenon Batang3, Abdullah H. Alshahri1, Seneshaw Tsegaye5, Michael Hegy54, Thomas M. Missimer5 1 King Abdullah University of Science and Technology (KAUST),

[Figure]

Water Desalination and Reuse Center (WDRC), Biological and Environmental Science and Engineering (BESE), Thuwal 23955-6900, Saudi Arabia 2Woods Hole Oceanographic Institution, Biology Department, Woods Hole, MA 02543, USA 3Coastal and Marine Resources Core Laboratory, King Abdullah University of Science and Technology (KAUST), Thuwal, Saudi Arabia 4Guangzhou Institute of Advanced Technology, CAS, Haibin Road #1121, Nansha district, Guangzhou 511458, China 5U. A. Whitaker College of Engineering, Emergent Technologies Institute, Florida Gulf Coast University, 16301 Innovation Lane, Fort Myers, Florida 33965-6565 6Desalination Technologies Research Institute (DTRI), Saline Water Conversion Corporation (SWCC), P.O. Box 8328, Al-Jubail 31951, Saudi Arabia

Abstract: Binding of particulate and dissolved organic matter in the water column by marine gels allows sinking and cycling of organic matter into deeper water of the Red Sea and other marine water bodies. A series of four offshore profiles were made at which concentrations of bacteria, algae, particulate transparent exopolymer particles (p-TEP), colloidal transparent exopolymer particles (c-TEP), and the fractions of natural organic matter (NOM), including biopolymers, humic substances, low molecular weight neutrals, and low molecular weight acids were measured to depths ranging from 90 to 300 m. It was found that a statistically-significant relationship occurs between the concentrations of p-TEP withand bacteria and algae, but not with TOC in the offshore profiles, but not the nearshore samples.
[revised manuscript text omitted]
. 2). Note that the calibration curve for samples collected at sites A, B, and C are shown in Figure 2a and the curve for site D is shown as Figure 2b. The TOC concentrations of xanthan gum before and after 0.4 $\mu$m filtration were analyzed, and the TOC concentration difference was used to calculate the gum mass on each filter and the TEP concentration was estimated using the calibration curve. The same procedures were used for the 0.1 $\mu$m membrane to establish the calibration curve for colloidal particles. Afterwards, the TEP concentration was expressed in terms of Xanthan Gum equivalent in $\mu$g Xeq./L by dividing the TEP mass by the corresponding volume of TEP samples. Because particulate and colloidal TEP is determined indirectly, these values must be considered to be semi-quantitative. The new method developed by Villacorte et al. (2009) for TEP measurement was not used, as it would limit the comparability of the measured data with previous results.

2.6 Statistical methods used for data comparison

It is essential to perform a multidimensional regression analysis at a certain meaningful abstraction level to find interesting patterns and to determine whether the result of a data set is statistically significant (Chen et al., 2002). Multiple regression has been used by several researchers and practitioners for theory testing or explanation purposes. The question of interest becomes understanding the significance of the variables, and the variation and interaction between them (Tonidandel and LeBreton,

2011). It is herein desired to highlight the correlation between TEP concentrations and abundance of microalgae, bacteria, TOC, and dissolved fractions of NOM, including biopolymers. Thus, a multi-dimensional regression analysis, a correlation matrix, and two-way analysis of variance (ANOVA) with replication were performed to test the interaction and statistical significance between the various organic properties. In order to perform multiple regression analysis, there must be a relationship between the outcome variable and the independent variables the residuals are normally distributed, and the independent variables should not be highly correlated with each other (Osborne and Waters, 2002; Williams et al., 2013). The interdependency was tested using Pearson's bivariate correlations matrix. The spatial/temporal variation of the data is not the focus of this paper and the intricacies of dominance or relative weight between variables were not considered. The correlation coefficients, R2 and p-values were calculated to assess the relationship, statistical significance and interaction between various organic properties. The correlation matrix is an identity matrix, which would indicate that variables are related or unrelated. Multi regression analysis is used to understand the significance of the two or more organic properties in predicting the value of a criterion variables (TEP and Biopolymers). The two-way ANOVA analyses were used to determine the significant difference and interaction among the sites and organic properties. 
[revised manuscript text omitted]
 (28-164 $\mu$g/L, 42, 62 $\mu$g/L), humic substances (159-442 $\mu$g/L, 42, 248 $\mu$g/L), building blocks (81-260 $\mu$g/L, 42, 118 $\mu$g/L), LMW neutrals (16-477 $\mu$g/L, 42, 271 $\mu$g/L), and LMW acids (10-130 $\mu$g/L, 42, 40 $\mu$g/L). The range in biopolymer concentrations in the surface offshore samples are similar to the nearshore samples. All of the NOM fractions have higher concentrations at the A, B, and C profiles compared to the deep profile. The biopolymer fraction of NOM shows

a general reduction with depth in all offshore profiles. At sites A and B there is a spike in biopolymers at 10 m with minor variation between 10 m to 90 m. In the deeper profile, there is considerable variation in the photic zone with the surface having the highest value and subsequent spikes occurring at 30 and 60 m. Beginning at about 90 m, Tthere is a constant downward trend in concentration beginning at about 90 m. Humic acid concentrations showed only minor variations with depth in the shallow profiles, but the deep profile showed a reduction by about 29% from 90 to 300 m depth. There is a general decreasing trend in concentration of building blocks with depth at the deep site and only minimal differences throughout the depth profiles at sites A−C (Figs. 7 and 8). The concentrations of LMW neutrals at the shallow sites were the highest amongst NOM fractions, although with a wide range of variation. In contrast, LMW acids had the lowest concentrations without marked discrepancies in concentration in the vertical profiles between sites A−C, but a general reduction occurred below 120 m in the deep profile (Figs. 7 and 8).

**4 Discussion**

**4.1 Algal and cyanobacterial concentrations**

The flow cytometry approach used in this study was usedhighly effective in characterizing and enumerating the small size classes of phytoplankton and cyanobacteria that are readily distinguishable on the basis of cell size and autofluorescence. Thus, cyanobacteria (presumably Synechococcus spp), Prochlorococcus, and the general class of pico/nanoplankton were numerically dominant, with very few larger eukaryotic algal species detected. This is consistent with prior studies that reported that phytoplankton in the oligotrophic northern Red Sea and Gulf of Aqaba are dominated (>95%) by cells <5 $\mu$m in size (Lindell and Post 1995; Yahel et al. 1998). Only during the summer does the large macroalgae Trichodesmium sp. also become prominent. As reported here, algae ranging from 5 to several hundred $\mu$m are extremely scarce, although not totally absent (Sommer 2000; Kimor and Goldanski 1992).

**4.2 Statistical significance and dependency of p-TEP, c-TEP and biopolymers**

The two-way ANOVA was employed to provide an important insight into the pattern of the data and its interdependency. Each organic parameter (sample) has been drawn independently of the other parameters and is normally distributed. The analysis shows that there is a significant difference ($p<0.05$) in the mean of the concentrations of TEP, bacteria, algae, TOC and biopolymer, and the mean of the sites. Also, it shows that there is no interdependency between the sites where samples are measured (Table 2). In order to validate the appropriateness of the multiple regression analysis, multi-collinearity of the concentration of Bacteria, Algae, Biopolymer, TOC, p-TEP, and c-TEP were checked using a bivariate correlation matrix (Table 3). The matrix of Pearson's bivariate correlations among all independent variables shows that the magnitude of the correlation coefficients are less than 0.8. A series of multi regression statistical analyses were preformed to test if there are significant relationships between dependent and independent variables/ organic properties. A summary of results of this analysis is presented and shown in Table 3.

4.2.1 p-TEP Multiple regression analysis between the p-TEP and the concentrations of bacteria, and algae, shows a significant statistical correlation for all offshore profiles while the concentrations of TOC is not significant parameter for the p-TEP. The overall regression is significant when the three variables are considered as a group. The individual variables and their significance are shown in Tables 3 and 4. However, the analysis shows that there was no statistically-significant relationship between p-TEP and the three parameters in the shallow, nearshore samples (Tables 3 and 4).

4.2.2 c-TEP The overall regression between the c-TEP and the concentrations of bacteria, algae, and TOC shows no significant statistical correlation for the shallow, nearshore (sites A, B, C) and offshore (Site D) profiles. However, the individual linear regression analysis shows statistically-significant relationships between c-TEP and TOC, and c-TEP and bacteria. There is no statistical relationship of significance between p-TEP and c-TEP with an adjusted $R^2=0.12$. Also, the nearshore samples show

that there was no statistically-significant relationship between c-TEP and TOC and Bacteria (Tables 3 and 4).

4.2.3 Biopolymers Multiple regression analysis between the biopolymers and the concentrations of bacteria, p-TEP and c-TEP shows a significant statistical correlation for all the offshore profiles while the concentration of algal and cyanobacteria is not a significant predictor of the biopolymers. The overall regression is significant when the four variables are considered as a group. The individual variables and their significance are shown in Table 3. The relationship between the biopolymers and all the four independent variables shows no significant statistical correlation in the shallow, nearshore samples.

4.32 Correlations between TEP, bacteria, algae, the biopolymer fraction of NOM, and TOC

TEP is composed of acidic polysaccharides and some large proteins that occur mostly in the biopolymer fraction of NOM and some of the proteins within the humic acid part of NOM (Bar-Zeev et al. 2015; Winters et al. 2016). TEP can be produced both abiotically and as extracellular discharges from bacteria and algae (Zhou et al. 1998; Passow et al. 2001; Passow 2002; Engel et al. 2004; Iuculano et al. 2017). Therefore, there should be some statistical relationship between TEP, the biopolymer fraction of NOM, bacterial concentration or algal concentration. A series of statistical analyses were preformed to test if there are significant relationships between the various organic properties (Tables 2, 3 and 4). There is a significant statistical relationship between p-TEP with algae (grouped with cyanobacteria) and bacteria but this does not occur with c-TEP. There is a significant statistical relationship between the biopolymer fraction of NOM with bacteria, p-TEP, and c-TEP in all of the offshore profiles. Since c-TEP is considered to be the precursor to formation of p-TEP, the association with biopolymers is logical and could indicate potential for abiotic assembly in the water column. 
[revised manuscript text omitted]
. The multiple regression analysis showed that p-TEP is correlated with bacteria and algae offshore, but no statistical correlation occurs between these variable in the nearshore data. No statistical correlation occurs between c-TEP and bacteria, algae, and TOC in offshore and nearshore samples. Also, p-TEP and c-TEP are not statistically correlated. The concentrations of biopolymers is correlated to bacteria, p-TEP, and c-TEP in the offshore samples, but the onshore samples. The relationships between p-TEP and c-TEP and other organic parameters, especially the biopolymer fraction of NOM, is different when comparing the offshore water column to the nearshore area. Seasonal differences during sampling could have also impacted the results. The only statistically-significant relationship in the measured parameters in the nearshore was that between bacteria and TOC. Differences Irregularity in local conditions, such as circulation and anthropogenic influences, in the nearshore zone can causecauses large variations in the organic parameters measured, not allowing statistically-significant relationships to be established.

References

[revised manuscript text omitted]

Source of Variation P-value Sample (offshore and nearshore site) 0.00702 Attributes (bacteria, algae, TOC, biopolymer and TEP 0.00000 Interaction 0.00011

Table 3. Correlation matrix

P-TEP ($\mu$g Xeq./L) C-TEP ($\mu$g Xeq./L) Bacterial count (cells/ml) Total algae (events/ml) TOC (mg/L) Biopolymers P-TEP ($\mu$g Xeq./L) 1.000 C-TEP ($\mu$g Xeq./L) 0.327 1.000 Bacteria (cells/ml) 0.152 0.075 1.000 Slgae (events/ml) 0.320 -0.019 0.016 1.000 TOC (mg/L) 0.259 0.422 0.374 -0.169 1.000 Biopolymers -0.263 0.207 -0.006 -0.304 0.209 1.000

Table 4. Multiple Regression analysis of selected organic parameters at the 0.05 significance level

Dependent variable Location Attributes P-value R Square Adjusted R Square Overall Significance p-TEP Offshore Bacterial count (cells/ml) 0.00106 0.51941 0.48874 0.0000001 Total algae (events/ml) 0.00127 TOC (mg/L) 0.76006 Nearshore Bacterial count (cells/ml) 0.48082 0.13834 0.01525 0.36195 Total algae (events/ml) 0.15212

TOC (mg/L) 0.37855

c-TEP Offshore Bacterial count (cells/ml) 0.91115 0.29356 0.24847 0.00090 Total algae (events/ml) 0.93401 TOC (mg/L) 0.00540 Nearshore Bacterial count (cells/ml) 0.88360 0.13412 0.01042 0.37736 Total algae (events/ml) 0.37529 TOC (mg/L) 0.18425

c-TEP Offshore p-TEP ($\mu$g Xeq./L) 0.00750 0.13704 0.11943 0.00750 Nearshore p-TEP ($\mu$g Xeq./L) 0.06876 0.14271 0.10374 0.06876

Biopolymers offshore Bacterial count (cells/ml) 0.00296 0.36179 0.32106 0.00009 P-TEP ($\mu$g Xeq./L) 0.00001 C-TEP ($\mu$g Xeq./L) 0.03707 Nearshore Bacterial count (cells/ml) 0.07978 0.21256 0.05507 0.28648 Total algae (events/ml) 0.51524 P-TEP ($\mu$g Xeq./L) 0.38547 C-TEP ($\mu$g Xeq./L) 0.34150

Acknowledgments

Funding for the offshore sample collection was provided by the King Abdullah University of Science and Technology Coastal and Marine Resources Core Laboratory. Analytical work was funded by the Water Desalination and Reuse Center, King Abdullah University of Science and Technology. Support for DMA was provided by the National Science Foundation (Grants OCE-0850421 OCE-0430724, OCE-0911031, and OCE-1314642) and National Institutes of Health (NIEHS-1P50-ES021923-01) through the Woods Hole Center for Oceans and Human Health.

Conflicts of Interest None declared

Figure captions

Fig. 1. Map showing the sampling profile locations in the Red Sea

Fig. 2. Xanthan gum standard calibration curves for determination of p-TEP and c-TEP. Curve a is for sites A, B, and C and curve b is for site D.

Fig. 3. Physical data from the three 90 m profiles

Fig. 4. Physical data from the 300 m profile

Fig. 5. Algal and cyanobacteria composition and concentration data from the three 90 m profiles

Fig. 6. Algal and cyanobacteria composition and concentration data from the 300 m profile

Fig. 7. Organic carbon concentrations for the three 90 m profiles

Fig. 8. Organic carbon concentrations from the 300 m profile

---

## Author Comment (AC4) · 25 Jul 2019

eTTransparent exopolymer particle occurrence and interaction with algae, bacteria, and the fractions of organic carbonbinding of organic and inorganic particles in the Red Sea: Implications for downward transport of biogenic materials

Abdullah H. A. Dehwah1,6, Donald M. Anderson2, Sheng Li1,4, Francis L. Mallon3, Zenon Batang3, Abdullah H. Alshahri1, Seneshaw Tsegaye5, Michael Hegy54, Thomas M. Missimer5 1 King Abdullah University of Science and Technology (KAUST),

[Figure]

Water Desalination and Reuse Center (WDRC), Biological and Environmental Science and Engineering (BESE), Thuwal 23955-6900, Saudi Arabia 2Woods Hole Oceanographic Institution, Biology Department, Woods Hole, MA 02543, USA 3Coastal and Marine Resources Core Laboratory, King Abdullah University of Science and Technology (KAUST), Thuwal, Saudi Arabia 4Guangzhou Institute of Advanced Technology, CAS, Haibin Road #1121, Nansha district, Guangzhou 511458, China 5U. A. Whitaker College of Engineering, Emergent Technologies Institute, Florida Gulf Coast University, 16301 Innovation Lane, Fort Myers, Florida 33965-6565 6Desalination Technologies Research Institute (DTRI), Saline Water Conversion Corporation (SWCC), P.O. Box 8328, Al-Jubail 31951, Saudi Arabia

Abstract: Binding of particulate and dissolved organic matter in the water column by marine gels allows sinking and cycling of organic matter into deeper water of the Red Sea and other marine water bodies. A series of four offshore profiles were made at which concentrations of bacteria, algae, particulate transparent exopolymer particles (p-TEP), colloidal transparent exopolymer particles (c-TEP), and the fractions of natural organic matter (NOM), including biopolymers, humic substances, low molecular weight neutrals, and low molecular weight acids were measured to depths ranging from 90 to 300 m. It was found that a statistically-significant relationship occurs between the concentrations of p-TEP withand bacteria and algae, but not with TOC in the offshore profiles, but not the nearshore samples.
[revised manuscript text omitted]
. 2). Note that the calibration curve for samples collected at sites A, B, and C are shown in Figure 2a and the curve for site D is shown as Figure 2b. The TOC concentrations of xanthan gum before and after 0.4 $\mu$m filtration were analyzed, and the TOC concentration difference was used to calculate the gum mass on each filter and the TEP concentration was estimated using the calibration curve. The same procedures were used for the 0.1 $\mu$m membrane to establish the calibration curve for colloidal particles. Afterwards, the TEP concentration was expressed in terms of Xanthan Gum equivalent in $\mu$g Xeq./L by dividing the TEP mass by the corresponding volume of TEP samples. Because particulate and colloidal TEP is determined indirectly, these values must be considered to be semi-quantitative. The new method developed by Villacorte et al. (2009) for TEP measurement was not used, as it would limit the comparability of the measured data with previous results.

2.6 Statistical methods used for data comparison

It is essential to perform a multidimensional regression analysis at a certain meaningful abstraction level to find interesting patterns and to determine whether the result of a data set is statistically significant (Chen et al., 2002). Multiple regression has been used by several researchers and practitioners for theory testing or explanation purposes. The question of interest becomes understanding the significance of the variables, and the variation and interaction between them (Tonidandel and LeBreton,

2011). It is herein desired to highlight the correlation between TEP concentrations and abundance of microalgae, bacteria, TOC, and dissolved fractions of NOM, including biopolymers. Thus, a multi-dimensional regression analysis, a correlation matrix, and two-way analysis of variance (ANOVA) with replication were performed to test the interaction and statistical significance between the various organic properties. In order to perform multiple regression analysis, there must be a relationship between the outcome variable and the independent variables the residuals are normally distributed, and the independent variables should not be highly correlated with each other (Osborne and Waters, 2002; Williams et al., 2013). The interdependency was tested using Pearson's bivariate correlations matrix. The spatial/temporal variation of the data is not the focus of this paper and the intricacies of dominance or relative weight between variables were not considered. The correlation coefficients, R2 and p-values were calculated to assess the relationship, statistical significance and interaction between various organic properties. The correlation matrix is an identity matrix, which would indicate that variables are related or unrelated. Multi regression analysis is used to understand the significance of the two or more organic properties in predicting the value of a criterion variables (TEP and Biopolymers). The two-way ANOVA analyses were used to determine the significant difference and interaction among the sites and organic properties. 
[revised manuscript text omitted]
 (28-164 $\mu$g/L, 42, 62 $\mu$g/L), humic substances (159-442 $\mu$g/L, 42, 248 $\mu$g/L), building blocks (81-260 $\mu$g/L, 42, 118 $\mu$g/L), LMW neutrals (16-477 $\mu$g/L, 42, 271 $\mu$g/L), and LMW acids (10-130 $\mu$g/L, 42, 40 $\mu$g/L). The range in biopolymer concentrations in the surface offshore samples are similar to the nearshore samples. All of the NOM fractions have higher concentrations at the A, B, and C profiles compared to the deep profile. The biopolymer fraction of NOM shows

a general reduction with depth in all offshore profiles. At sites A and B there is a spike in biopolymers at 10 m with minor variation between 10 m to 90 m. In the deeper profile, there is considerable variation in the photic zone with the surface having the highest value and subsequent spikes occurring at 30 and 60 m. Beginning at about 90 m, Tthere is a constant downward trend in concentration beginning at about 90 m. Humic acid concentrations showed only minor variations with depth in the shallow profiles, but the deep profile showed a reduction by about 29% from 90 to 300 m depth. There is a general decreasing trend in concentration of building blocks with depth at the deep site and only minimal differences throughout the depth profiles at sites A−C (Figs. 7 and 8). The concentrations of LMW neutrals at the shallow sites were the highest amongst NOM fractions, although with a wide range of variation. In contrast, LMW acids had the lowest concentrations without marked discrepancies in concentration in the vertical profiles between sites A−C, but a general reduction occurred below 120 m in the deep profile (Figs. 7 and 8).

4 Discussion

4.1 Algal and cyanobacterial concentrations

The flow cytometry approach used in this study was usedhighly effective in characterizing and enumerating the small size classes of phytoplankton and cyanobacteria that are readily distinguishable on the basis of cell size and autofluorescence. Thus, cyanobacteria (presumably Synechococcus spp), Prochlorococcus, and the general class of pico/nanoplankton were numerically dominant, with very few larger eukaryotic algal species detected. This is consistent with prior studies that reported that phytoplankton in the oligotrophic northern Red Sea and Gulf of Aqaba are dominated (>95%) by cells <5 $\mu$m in size (Lindell and Post 1995; Yahel et al. 1998). Only during the summer does the large macroalgae Trichodesmium sp. also become prominent. As reported here, algae ranging from 5 to several hundred $\mu$m are extremely scarce, although not totally absent (Sommer 2000; Kimor and Goldanski 1992).

**4.2 Statistical significance and dependency of p-TEP, c-TEP and biopolymers**

The two-way ANOVA was employed to provide an important insight into the pattern of the data and its interdependency. Each organic parameter (sample) has been drawn independently of the other parameters and is normally distributed. The analysis shows that there is a significant difference ($p < 0.05$) in the mean of the concentrations of TEP, bacteria, algae, TOC and biopolymer, and the mean of the sites. Also, it shows that there is no interdependency between the sites where samples are measured (Table 2). In order to validate the appropriateness of the multiple regression analysis, multi-collinearity of the concentration of Bacteria, Algae, Biopolymer, TOC, p-TEP, and c-TEP were checked using a bivariate correlation matrix (Table 3). The matrix of Pearson's bivariate correlations among all independent variables shows that the magnitude of the correlation coefficients are less than 0.8. A series of multi regression statistical analyses were preformed to test if there are significant relationships between dependent and independent variables/ organic properties. A summary of results of this analysis is presented and shown in Table 3.

4.2.1 p-TEP Multiple regression analysis between the p-TEP and the concentrations of bacteria, and algae, shows a significant statistical correlation for all offshore profiles while the concentrations of TOC is not significant parameter for the p-TEP. The overall regression is significant when the three variables are considered as a group. The individual variables and their significance are shown in Tables 3 and 4. However, the analysis shows that there was no statistically-significant relationship between p-TEP and the three parameters in the shallow, nearshore samples (Tables 3 and 4).

4.2.2 c-TEP The overall regression between the c-TEP and the concentrations of bacteria, algae, and TOC shows no significant statistical correlation for the shallow, nearshore (sites A, B, C) and offshore (Site D) profiles. However, the individual linear regression analysis shows statistically-significant relationships between c-TEP and TOC, and c-TEP and bacteria. There is no statistical relationship of significance between p-TEP and c-TEP with an adjusted $R^2 = 0.12$. Also, the nearshore samples show

that there was no statistically-significant relationship between c-TEP and TOC and Bacteria (Tables 3 and 4).

4.2.3 Biopolymers Multiple regression analysis between the biopolymers and the concentrations of bacteria, p-TEP and c-TEP shows a significant statistical correlation for all the offshore profiles while the concentration of algal and cyanobacteria is not a significant predictor of the biopolymers. The overall regression is significant when the four variables are considered as a group. The individual variables and their significance are shown in Table 3. The relationship between the biopolymers and all the four independent variables shows no significant statistical correlation in the shallow, nearshore samples.

4.32 Correlations between TEP, bacteria, algae, the biopolymer fraction of NOM, and TOC

TEP is composed of acidic polysaccharides and some large proteins that occur mostly in the biopolymer fraction of NOM and some of the proteins within the humic acid part of NOM (Bar-Zeev et al. 2015; Winters et al. 2016). TEP can be produced both abiotically and as extracellular discharges from bacteria and algae (Zhou et al. 1998; Passow et al. 2001; Passow 2002; Engel et al. 2004; Iuculano et al. 2017). Therefore, there should be some statistical relationship between TEP, the biopolymer fraction of NOM, bacterial concentration or algal concentration. A series of statistical analyses were preformed to test if there are significant relationships between the various organic properties (Tables 2, 3 and 4). There is a significant statistical relationship between p-TEP with algae (grouped with cyanobacteria) and bacteria but this does not occur with c-TEP. There is a significant statistical relationship between the biopolymer fraction of NOM with bacteria, p-TEP, and c-TEP in all of the offshore profiles. Since c-TEP is considered to be the precursor to formation of p-TEP, the association with biopolymers is logical and could indicate potential for abiotic assembly in the water column. 
[revised manuscript text omitted]
. The multiple regression analysis showed that p-TEP is correlated with bacteria and algae offshore, but no statistical correlation occurs between these variable in the nearshore data. No statistical correlation occurs between c-TEP and bacteria, algae, and TOC in offshore and nearshore samples. Also, p-TEP and c-TEP are not statistically correlated. The concentrations of biopolymers is correlated to bacteria, p-TEP, and c-TEP in the offshore samples, but the onshore samples. The relationships between p-TEP and c-TEP and other organic parameters, especially the biopolymer fraction of NOM, is different when comparing the offshore water column to the nearshore area. Seasonal differences during sampling could have also impacted the results. The only statistically-significant relationship in the measured parameters in the nearshore was that between bacteria and TOC. Differences Irregularity in local conditions, such as circulation and anthropogenic influences, in the nearshore zone can causecauses large variations in the organic parameters measured, not allowing statistically-significant relationships to be established.

References

[revised manuscript text omitted]

Location Date Depth (m) Total Algae (cells/mL) Bacteria (cells/mL) TOC (mg/L) NOM ($\mu$g/L) TEP $\mu$g Xeq./L Biopoly. Humic substances Building Blocks LMWN LWMA p-TEP c-TEP 1N. Obhor 1/8/2014 Surface 30,524 112,790 0.89 76 345 103 168 88 162 - 1Corniche 1/11/2014 Surface 3,603 196,377 0.94 90 360 91 192 94 121 73 1S. Jeddah 1/9/2014 Surface 1,677 264,728 1.02 116 351 139 197 103 157 122 1Buhayrat - Sur-

face 30,395 320,870 1.053 47 343 82 16 85 58 - 2Site A (Buhayrat) 1/7/2014 Surface 14,956 179,837 0.88 63 367 131 230 130 123 130 2Site B (Saudia) 5/25/2013 Surface 23,773 317,174 0.83 84 289 101 45 101 53 56 3N. Obhor 10/25/2014 Surface 129,738 520,350 1.1 57 205 95 163 18 318 90 3Corniche 11/6/2014 Surface 89,033 254,450 1.0 44 201 86 249 16 249 120 3S. Jeddah 12/24/2014 Surface 42,923 216,400 0.9 32 196 95 276 24 255 115 4N. Obhor 6/7/2015 Surface - 707,100 1.262 40 194 85 466 19 111 223 4N. Obhor 6/17/2015 Surface - - 1.034 42 185 99 231 18 - - 4N. Obhor 7/1/2015 Surface 108,740 282,450 1.162 49 192 105 313 21 142 189 4N. Obhor 7/12/2015 Surface 87,615 252,233 1.036 50 188 105 477 31 - - 4N. Obhor 8/3/2015 Surface 135,603 908,100 1.104 80 209 122 269 31 231 216 4N. Obhor 8/16/2015 Surface 49,770 1,764,850 1.118 71 184 111 284 16 - - 4Saudia 6/7/2015 Surface - 317,567 1.055 29 172 100 369 21 215 242 4Saudia 6/17/2015 Surface - - 1.233 46 189 84 183 14 - - 4Saudia 7/1/2015 Surface 61,925 583,400 1.287 44 190 93 152 14 146 213 4Saudia 7/12/2015 Surface 137,363 1,070,400 1.294 40 159 82 188 16 - - 4Saudia 8/3/2015 Surface 53,810 1,736,450 1.164 93 180 111 238 19 347 287 4Saudia 8/16/2015 Surface 43,060 2,182,550 1.181 83 208 103 276 15 5N. Ohbor 2015/4/2 Surface 91,870 1,356,600 1.10 55 214 98 387 14 261 132 6KAUST SW 5/3/2014 Surface 4,766 273,400 1.42 29 217 119 315 24 278 100 6KAUST SW 5/22/2014 Surface 9,350 236,000 1.037 55 197 121 252 17 346 97 6KAUST SW 6/11/2014 Surface 3,140 287,850 0.992 36 246 81 319 18 229 170 6KAUST SW 7/3/2014 Surface 4,958 324,600 1.085 43 212 151 227 10 85 127 6KAUST SW 7/19/2014 Surface 11,080 389,450 0.97 53 212 91 233 16 99 117 6KAUST SW 8/18/2014 Surface 6,057 316,450 1.112 40 201 88 225 14 82 112 6KAUST SW 9/18/2014 Surface 52,453 321,250 0.923 35 193 93 171 13 97 69 6KAUST SW 10/21/2014 Surface 12,228 630,600 0.831 39 193 108 256 16 213 50 6KAUST SW 12/3/2-14 Surface 10,673 347,133 1.004 33 189 101 288 23 138 43 6KAUST SW 2/11/2015 Surface 12,890 292,500 1.275 36 200 102 343 31 182 87 6KAUST SW 5/21/2015 Surface 28,009 450,800 0.93 31 177 93 236 25 143 50 6KAUST SW 8/6/2015 Surface 44,153 336,900 1.041 42 184 86 229 18 186 36 6KAUST SW 9/17/2015 Surface 52,453 297,867 1.084 28 188 86 230

24 300 105 7KAUST SW 9/30/2016 Surface 11,955 369,300 1.073 112 429 213 385 86 - - 7KAUST SW 10/2/2014 Surface 10,600 367,000 0.993 105 363 193 353 92 307 - 7KAUST SW 10/9/2014 Surface 17,777 368,463 0.944 140 373 218 335 79 333 - 7KAUST SW 10/16/2014 Surface 22,030 319,950 0.961 88 340 216 346 80 206 - 7KAUST SW 10/27/2014 Surface 42,550 297,700 0.917 164 348 260 468 49 318 - 7KAUST SW 11/6/2014 Surface 86,033 587,200 0.864 73 442 93 470 73 124 - 7KAUST SW 11/17/2014 Surface 107,030 673,700 0.897 71 374 221 352 81 83 - No. Samples 38 40 42 42 42 42 42 42 35 27 Range in values 1,677-137,363 112,790-2,182,550 0.830-1.420 28-164 159-442 81-260 16-477 10-130 53-347 36-287 Average 44,383 525,820 1 62 248 118 271 40 191 125 1 R. Rachman et al. 2015; 2 Dehwah et al. 2015; 3Dehwah and Missimer 2016, 4Alsahri et al. 2017, 5Dehwah et al 2017; 6Dehwah and Missimer 2017; 7Dehwah and Missimer 2015d Table 2. Regression analysis of selected organic parameters at the 0.05 significance level 0.05 level Organic Parameters Location R2 p-value Significant (?) p-TEP v. Bacteria Site A 0.6677 0.0039 Y Site B 0.7295 0.001656 Y Site C 0.6691 0.00383 Y Deep Profile (300 m) 0.3034 0.009661 Y Nearshore 0.0593 0.158757 N p-TEP v. Algae Site A 0.1011 0.37063 N Site B 0.5363 0.016017 Y Site C 0.2463 0.144607 N Deep Profile (300 m) 0.1495 0.083384 N Nearshore 0.0169 0.471436 N c-TEP v. Bacteria Site A 0.6677 0.0039 Y Site B 0.6430 0.005265 Y Site C 0.2474 0.143485 N Deep Profile (300 m) 0.5512 0.000116 N Nearshore 0.2622 0.006329 N c-TEP v. Algae Site A 0.1011 0.37063 N Site B 0.2900 0.108267 N Site C 0.0141 0.743804 N Deep Profile (300 m) 0.5713 7.4E-05 Y Nearshore 0.1476 0.057986 N Biopolymers v. Bacteria Site A 0.8166 0.000335 Y Site B 0.6726 0.003663 Y Site C 0.6868 0.003043 Y Deep Profile (300 m) 0.7814 1.08E-07 Y Nearshore 0.0123 0.495799 N Biopolymers v. Algae Site A 0.5801 0.010465273 Y Site B 0.2918 0.10701 N Site C 0.2996 0.101512 N Deep Profile (300 m) 0.7078 1.77E-06 Y Nearshore 0.0107 0.537011 N Biopolymers v. p-TEP Site A 0.4890 0.024407 Y Site B 0.4824 0.0258132 Y Site C 0.4020 0.049006 Y Deep Profile (300 m) 0.1551 0.077318 N Nearshore 0.0808 0.09790 N Biopolymers v. c-TEP Site A 0.4890 0.024407 Y Site B 0.3696 0.062253 N Site C 0.2590 0.13302

N Deep Profile (300 m) 0.5883 4.97E-05 Y Nearshore 0.0331 0.364097 N p-TEP v. c-TEP Site A 1 0 Y Site B 0.362578 0.065466 N Site C 0.3798 0.057758 N Deep Profile (300 m) 0.1660 0.066765 N Nearshore 0.0491 0.266597 N p-TEP v. TOC Site A 0.6591 0.00434 Y Site B 0.2760 0.118919 N Site C 0.0979 0.378796 N Deep Profile (300 m) 0.3156 0.008046 Y Nearshore 0.0284 0.332963 N c-TEP vs. TOC Site A 0.6591 0.0043396 Y Site B 0.0431 0.565154 N Site C 0.0165 0.723942 N Deep Profile (300 m) 0.6698 5.79E-06 Y Nearshore 0.1995 0.019513 N Bacteria v. TOC Site A 0.7717 0.000822 Y Site B 0.2994 0.101653 N Site C 0.1294 0.307187 N Deep Profile (300 m) 0.7812 1.08E-07 Y Nearshore 0.1144 0.032827 Y Algae v. TOC Site A 0.0928 0.3922134 N Site B 0.4907 0.024064 N Site C 0.3188 0.089015 N Deep Profile (300 m) 0.6220 2.16E-05 Y Nearshore 0.0388 0.236167 N

Table 2. Two-way ANOVA p-value for interdependency of site and its attributes

Source of Variation P-value Sample (offshore and nearshore site) 0.00702 Attributes (bacteria, algae, TOC, biopolymer and TEP 0.00000 Interaction 0.00011

Table 3. Correlation matrix

P-TEP ($\mu$g Xeq./L) C-TEP ($\mu$g Xeq./L) Bacterial count (cells/ml) Total algae (events/ml) TOC (mg/L) Biopolymers P-TEP ($\mu$g Xeq./L) 1.000 C-TEP ($\mu$g Xeq./L) 0.327 1.000 Bacteria (cells/ml) 0.152 0.075 1.000 Slgae (events/ml) 0.320 -0.019 0.016 1.000 TOC (mg/L) 0.259 0.422 0.374 -0.169 1.000 Biopolymers -0.263 0.207 -0.006 -0.304 0.209 1.000

Table 4. Multiple Regression analysis of selected organic parameters at the 0.05 significance level

Dependent variable Location Attributes P-value R Square Adjusted R Square Overall Significance p-TEP Offshore Bacterial count (cells/ml) 0.00106 0.51941 0.48874 0.0000001 Total algae (events/ml) 0.00127 TOC (mg/L) 0.76006 Nearshore Bacterial count (cells/ml) 0.48082 0.13834 0.01525 0.36195 Total algae (events/ml) 0.15212

TOC (mg/L) 0.37855

c-TEP Offshore Bacterial count (cells/ml) 0.91115 0.29356 0.24847 0.00090 Total algae (events/ml) 0.93401 TOC (mg/L) 0.00540 Nearshore Bacterial count (cells/ml) 0.88360 0.13412 0.01042 0.37736 Total algae (events/ml) 0.37529 TOC (mg/L) 0.18425

c-TEP Offshore p-TEP ($\mu$g Xeq./L) 0.00750 0.13704 0.11943 0.00750 Nearshore p-TEP ($\mu$g Xeq./L) 0.06876 0.14271 0.10374 0.06876

Biopolymers offshore Bacterial count (cells/ml) 0.00296 0.36179 0.32106 0.00009 P-TEP ($\mu$g Xeq./L) 0.00001 C-TEP ($\mu$g Xeq./L) 0.03707 Nearshore Bacterial count (cells/ml) 0.07978 0.21256 0.05507 0.28648 Total algae (events/ml) 0.51524 P-TEP ($\mu$g Xeq./L) 0.38547 C-TEP ($\mu$g Xeq./L) 0.34150

Acknowledgments

Funding for the offshore sample collection was provided by the King Abdullah University of Science and Technology Coastal and Marine Resources Core Laboratory. Analytical work was funded by the Water Desalination and Reuse Center, King Abdullah University of Science and Technology. Support for DMA was provided by the National Science Foundation (Grants OCE-0850421 OCE-0430724, OCE-0911031, and OCE-1314642) and National Institutes of Health (NIEHS-1P50-ES021923-01) through the Woods Hole Center for Oceans and Human Health.

Conflicts of Interest None declared

Figure captions

Fig. 1. Map showing the sampling profile locations in the Red Sea

Fig. 2. Xanthan gum standard calibration curves for determination of p-TEP and c-TEP. Curve a is for sites A, B, and C and curve b is for site D.

Fig. 3. Physical data from the three 90 m profiles

Fig. 4. Physical data from the 300 m profile

Fig. 5. Algal and cyanobacteria composition and concentration data from the three 90 m profiles

Fig. 6. Algal and cyanobacteria composition and concentration data from the 300 m profile

Fig. 7. Organic carbon concentrations for the three 90 m profiles

Fig. 8. Organic carbon concentrations from the 300 m profile

[Figure]